# NLRP3 inflammasome in fibroblasts links tissue damage with inflammation in breast cancer progression and metastasis

Nour Ershaid [1,9], Yoray Sharon[1,9], Hila Doron[1], Yael Raz[1,2], Ophir Shani[1], Noam Cohen[1], Lea Monteran[1], Leonor Leider-Trejo[3], Amir Ben-Shmuel[4], Muhammad Yassin[1], Motti Gerlic [5], Adit Ben-Baruch[6], Metsada Pasmanik-Chor[7], Roni Apte[8] & Neta Erez[1]*

Cancer-Associated Fibroblasts (CAFs) were shown to orchestrate tumour-promoting inflammation in multiple malignancies, including breast cancer. However, the molecular pathways that govern the inflammatory role of CAFs are poorly characterised. In this study we found that fibroblasts sense damage-associated molecular patterns (DAMPs), and in response activate the NLRP3 inflammasome pathway, resulting in instigation of pro-inflammatory signalling and secretion of IL-1β. This upregulation was evident in CAFs in mouse and in human breast carcinomas. Moreover, CAF-derived inflammasome signalling facilitated tumour growth and metastasis, which was attenuated when NLRP3 or IL-1β were specifically ablated. Functionally, CAF-derived inflammasome promoted tumour progression and metastasis by modulating the tumour microenvironment towards an immune suppressive milieu and by upregulating the expression of adhesion molecules on endothelial cells. Our findings elucidate a mechanism by which CAFs promote breast cancer progression and metastasis, by linking the physiological tissue damage response of fibroblasts with tumour-promoting inflammation.

[1] Department of Pathology, Sackler Faculty of Medicine, Tel Aviv University, Tel Aviv, Israel. [2] Department of Obstetrics and Gynecology, Tel Aviv Sourasky Medical Center, Tel Aviv, Israel. [3] Department of Pathology, Tel Aviv Sourasky Medical Center, Tel Aviv University, Tel Aviv, Israel. [4] Department of Infectious Diseases, Israel Institute for Biological Research, Ness Ziona, Israel. [5] Department of Clinical Microbiology and Immunology, Sackler Faculty of Medicine, Tel Aviv University, Tel Aviv, Israel. [6] Department of Cell Research and Immunology, Tel Aviv University, Tel Aviv, Israel. [7] Bioinformatics Unit, George S. Wise Faculty of Life Science, Tel Aviv University, Tel Aviv, Israel. [8] The Shraga Segal Department of Microbiology, Immunology and Genetics, Ben Gurion University of the Negev, Beer-Sheva, Israel. [9] These authors contributed equally: Nour Ershaid, Yoray Sharon. *email: netaerez@tauex.tau.ac.il

Breast cancer remains one of the leading causes of cancer-related deaths, and the most frequently diagnosed cancer in women. The tumour microenvironment plays a crucial role in cancer growth and metastasis[1]. Cancer-associated fibroblasts (CAFs) are the most prominent cell type in the microenvironment of breast tumours, and their abundance is correlated with a higher grade of malignancy and with poor prognosis[2]. CAFs facilitate tumour progression by secreting growth-promoting factors, by enhancing angiogenesis and by remodelling the extracellular matrix (ECM)[3]. Inflammation is now established as a hallmark of cancer[4], and CAFs were shown to be key players in mediating tumour-promoting inflammation[5–7], due to their capability to recruit immune cells, and to affect the tumour-immune system interactions[6,8,9]. However, the triggers and the molecular signalling which regulate the inflammatory role of CAFs are still poorly characterised.

Inflammasomes are a family of multi-protein complexes, expressed by cells of the innate immune system as part of its pathogen pattern recognition and damage-sensing mechanisms[10]. The NLRP3 inflammasome functions in sensing of tissue damage during sterile inflammation by recognising self-derived danger signals termed damage-associated molecular patterns (DAMPs)[11,12]. DAMP recognition promotes the maturation and secretion of pro-inflammatory cytokines, including IL-1β, a pleiotropic pro-inflammatory cytokine that has profound effects on inflammation and immunity. During chronic inflammation that invokes extensive tissue damage and constant release of DAMPs, sustained activation of inflammasomes may enhance collateral damage; e.g. in rheumatoid arthritis[13], systemic sclerosis[14], and neurodegenerative diseases[15].

The microenvironment of tumours is burdened with DAMPs due to necrosis of cancer cells and persistent tissue damage, which were shown to induce activation of the inflammasome in immune cells[16]. However, the role of immune cell-derived inflammasome signalling in tumours is controversial, and has been shown to have both pro-tumourigenic and anti-tumourigenic activities in various carcinomas[17]. Activation of the NLRP3 inflammasome in fibroblasts was reported in myocardial dysfunction[18], and in fibrosis[19], but it was not previously implicated in CAFs.

In this study, we set out to characterise the dynamic co-evolution of CAFs during breast carcinogenesis. We found that the NLRP3 inflammasome pathway is upregulated in CAFs during spontaneous murine mammary carcinogenesis, as well as in CAFs in human breast cancer. Moreover, we demonstrate that fibroblasts can function as DAMP sensors, and respond to tissue damage by activation of the inflammasome and secretion of IL-1β. Notably, genetic ablation of CAF-derived NLRP3 or IL-1β in different syngeneic mouse models delayed mammary tumour growth and/or attenuated lung metastasis. Functionally, we show that CAF-derived IL-1β facilitated tumour progression and metastasis by upregulating the expression of adhesion molecules in endothelial cells (ECs), and promoting the recruitment of CD11b$^+$Gr1$^+$ myeloid-derived suppressor cells (MDSCs). Thus, our findings uncovered a role for CAFs in breast cancer that links tissue damage with tumour-promoting inflammation via the NLRP3 inflammasome.

## Results

### The NLRP3 inflammasome is upregulated in mammary CAFs.
In an effort to characterise the dynamic changes in CAFs during breast cancer progression, we profiled the transcriptome of fibroblasts isolated from distinct stages during mammary carcinogenesis, utilising the MMTV-PyMT murine model of breast cancer. This model is characterised by multistep carcinogenesis including hyperplasia at 4–5 weeks, carcinoma at 9–10 weeks, and

advanced carcinoma at 12–13 weeks[20]. Fibroblasts were isolated using fluorescence-activated cell sorting (FACS) as previously described[21], and unbiased transcriptome analysis was performed utilising affymetrix gene arrays (Fig. 1a). Importantly, isolated fibroblasts were used directly from fresh tissues and were not cultured prior to expression profiling. Analysis of the results revealed that fibroblasts exhibited unique gene expression signatures that corresponded to distinct tumourigenic stages (Fig. 1b). Interestingly, there was only partial overlap in gene expression between the stages characterised. CAFs isolated from tumours (carcinoma and advanced carcinoma) were more similar to each other than to fibroblasts isolated from normal mammary glands, or from mammary hyperplastic lesions. 1157 differentially expressed genes overlapped between CAFs isolated from tumours (carcinoma and advanced carcinoma), while only a total of 69 genes overlapped between fibroblasts isolated from preneoplastic lesions and CAFs (Fig. 1c). These results suggest that fibroblasts co-evolve with tumour progression, and that a unique molecular programme is initiated in breast CAFs.

Further analysis of specific pathways that were activated in mammary CAFs revealed activation of pro-inflammatory genes, in agreement with previous studies[5–7] (Fig. 1d). Remarkably, we found that multiple genes that belong to the NLRP3 inflammasome pathway, including *P2rx7*, *Nlrp3*, *Casp1*, *Il1a* and *Il1b* are upregulated in CAFs isolated from mammary carcinoma, but not in normal mammary tissue, or in fibroblasts isolated from mammary hyperplasia (Fig. 1d, e). Notably, *Il18*, which is also activated by the NLRP3 inflammasome[17], was downregulated in mammary CAFs.

The activity of the NLRP3 inflammasome in fibroblasts was previously reported in pathologies that involve tissue damage, such as myocardial dysfunction[18,22,23], oral lesions[24], and fibrosis[19]. However, the NLRP3 inflammasome was not previously implicated in CAFs. Intrigued by these observations, we decided to focus on characterising the NLRP3/IL-1β pathway in CAFs. Validation of the transcriptome profiling results by qRT-PCR in mammary CAFs isolated from mammary tumours or from normal mammary glands by FACS (Supplementary Fig. 1a) confirmed the upregulation of genes related to the NLRP3 inflammasome pathway (Fig. 1f). Additionally, immunohisto-chemistry (IHC) of advanced mammary carcinoma tissue sections from MMTV-PyMT mice confirmed that in addition to its activation in tumour cells, NLRP3 is upregulated in CAFs, but not in normal mammary fibroblasts (NMFs) (Fig. 1g, h, Supplementary Fig. 2a). Moreover, intracellular flow cytometry analysis validated the upregulation of NLRP3 in CAFs isolated from mammary tumours (Supplementary Fig. 2b,c). Interestingly, transcriptome analysis of normal dermal fibroblasts compared with CAFs isolated from squamous cell carcinoma confirmed that activation of the inflammasome is not limited to breast cancer (Supplementary Fig. 2d).

### The NLRP3 inflammasome is upregulated in human breast CAFs.
To assess if upregulation of the NLRP3 pathway in the stroma of breast tumours is relevant to human breast cancer, we analysed a publicly available data set of tumour stroma samples that were obtained by laser micro-dissection of breast tumours from 53 patients[25]. Analysis revealed a significant up-regulation in the expression of genes from the NLRP3 inflammasome pathway (*NLRP3*, *PYCARD*, *CASP1*, and *IL-1β*) in the stroma of human breast tumours compared with normal stroma (Fig. 2a–d), suggesting a functional role for this pathway in tumour progression.

Notably, analysis of pro-inflammatory genes that were upregulated in mammary CAFs using the STRING database of

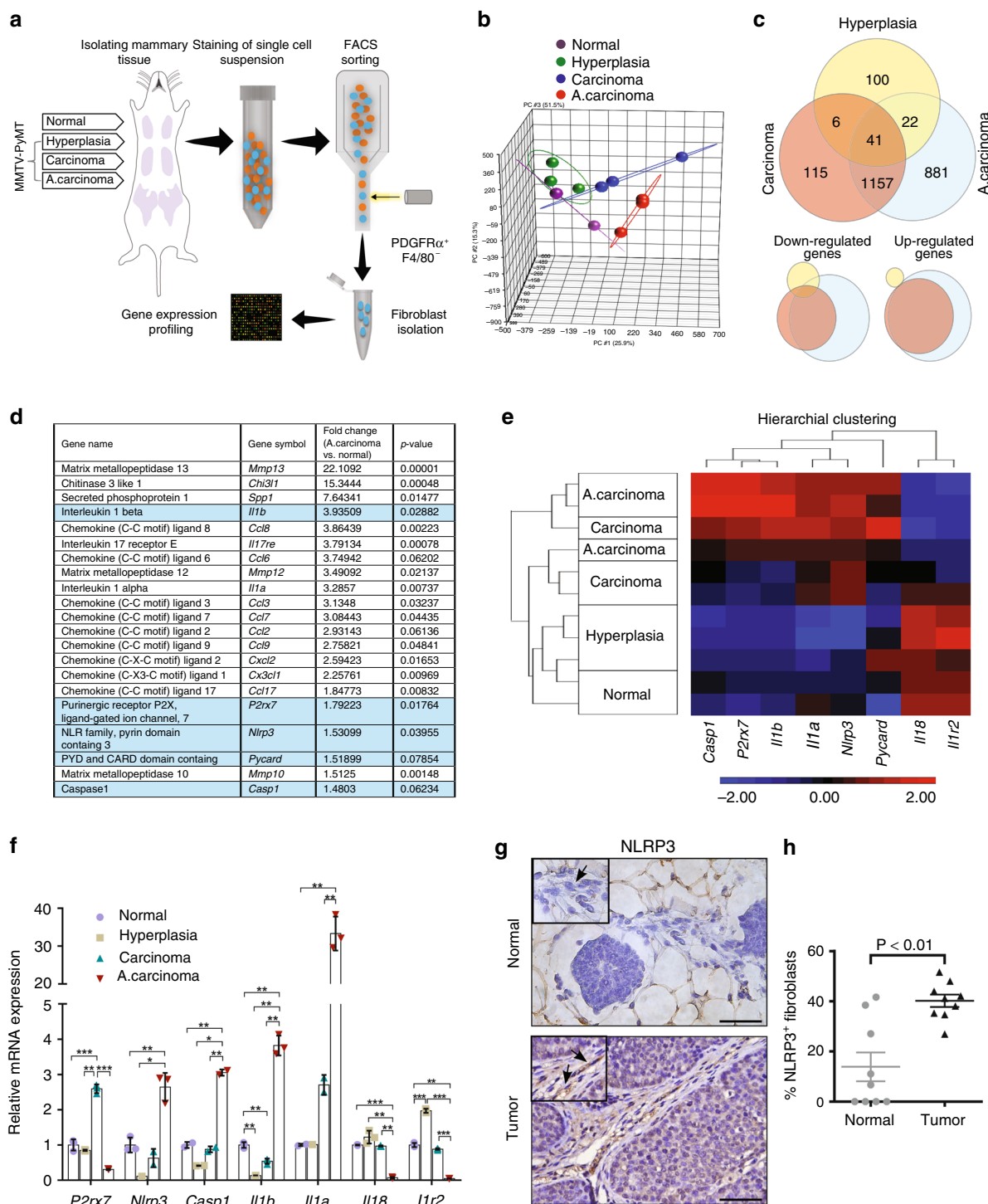

**Fig. 1** The NLRP3 inflammasome pathway is upregulated in fibroblasts during mammary carcinogenesis. **a** Scheme of fibroblast isolation procedure. **b** Principal component analysis (PCA) of samples included in the microarray transcriptome analysis. Each focal point represents a cohort of mice. Normal $n = 6$ mice/cohort, Hyperplasia $n = 6$ mice/cohort, Carcinoma $n = 4$ mice/cohort, advanced carcinoma (A. Carcinoma) $n = 3$ mice/cohort. **c** Venn diagrams depicting overlap of differentially expressed genes in fibroblasts isolated from distinct stages of MMTV-PyMT mammary carcinoma. **d** Pro-inflammatory genes upregulated in CAFs isolated from MMTV-PyMT tumours. Inflammasome-related genes are highlighted. **e** Heat map of *Nlrp3/Il1b* pathway-related genes. **f** qRT-PCR analysis of *Nlrp3/Il1b* pathway related genes in fibroblasts isolated by FACS (PDGFRα⁺CD45⁻EpCAM⁻) from MMTV-PyMT mice or normal mammary glands. Data are presented as mean ± s.d of technical repeats; One-way analysis of variance followed by Tukey's test. *$p <$ 0.05, **$p <$ 0.005, ***$p <$ 0.0005, $n =$ pools of 3 mice/group. Results are representative of three independent biological experiments. **g** Representative IHC staining of NLRP3 in PyMT tumours or in normal mammary glands. Arrows indicate fibroblasts. Scale bar, 25 μm. **h** Quantification of staining shown in **g**. Multiple fields of 3 mice/group were analysed for the percentage of NLRP3 expressing fibroblasts out of total fibroblasts. Data are presented as mean ± s.e. m; Mann–Whitney test. Source data are provided as a Source Data file

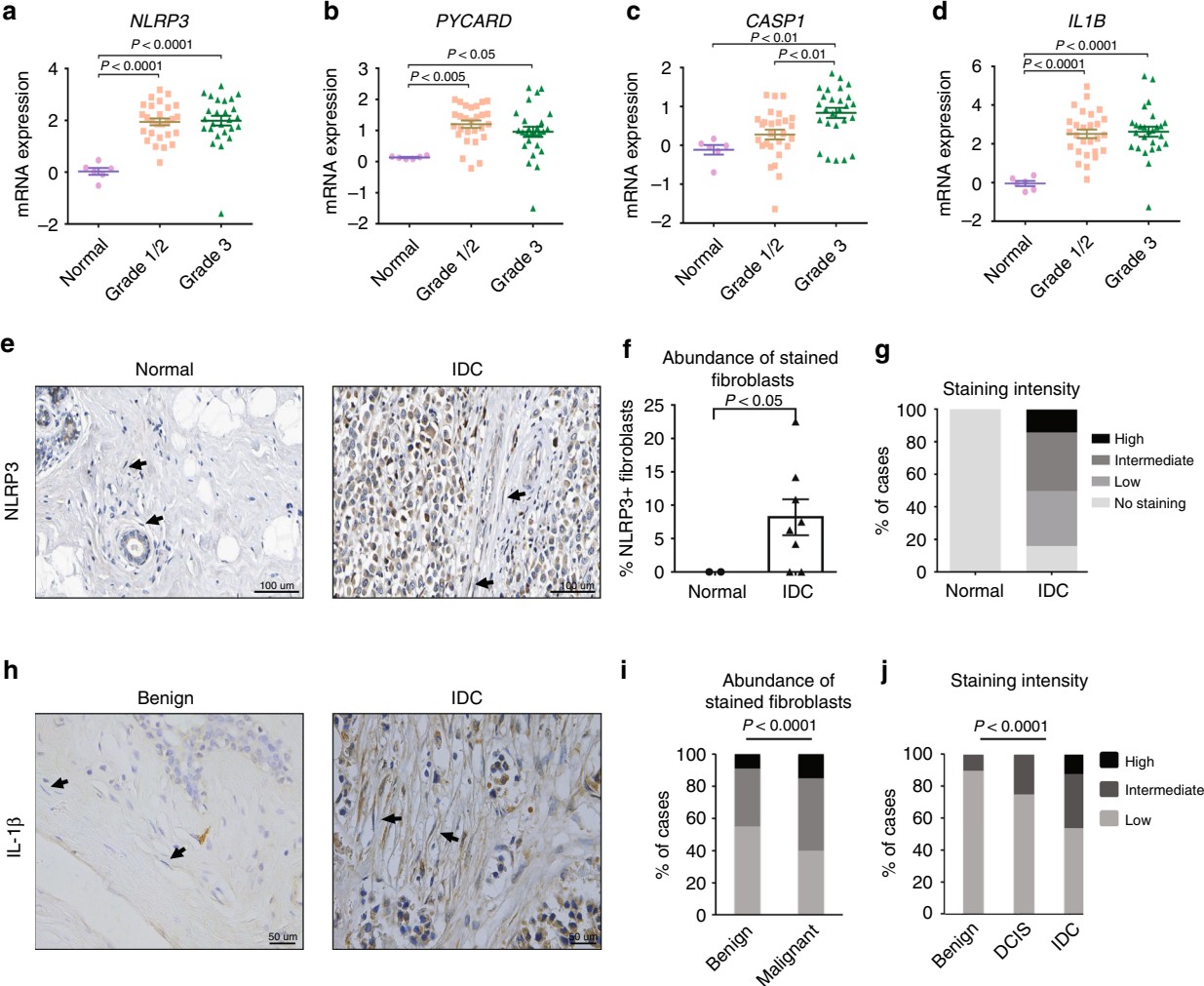

**Fig. 2** The NLRP3 inflammasome pathway is up-regulated in CAFs in human breast tumours. **a–d** Expression levels of *Nlrp3/Il1b* pathway-related genes in tumour-associated stroma of breast cancer patients with infiltrating ductal carcinoma (IDC) staged 1/2 vs. 3 ($n = 53$) or in normal breast stroma ($n = 6$). Data was obtained from NCBI GEO (Dataset accession number GSE 9014) and is presented as mean ± s.e.m.; One-way analysis of variance test followed by Tukey's multiple comparisons test. **e–g** Analysis of NLRP3 expression in fibroblasts in human IDC or in normal breast tissue sections. Images were obtained from the Human Protein Atlas. **e** Representative images, arrows in insets indicate fibroblasts. **f** Quantification of the percentage of NLRP3-expressing fibroblasts out of total fibroblasts for each sample. $n = 2$ samples for normal and 8 samples for IDC, error bars represent s.e.m; Welch's *t*-test. **g** Quantification of staining intensity for NLRP3 in fibroblasts shown in **e**. **h–j** Staining for IL-1β in benign or malignant (DCIS/IDC) tissue sections from human breast cancer patients. **h** Representative images of stained benign and IDC breast tissue sections. Arrows indicate fibroblasts. Scale bar, 50 μm. **i**, **j** Quantification of the abundance **i** and staining intensity **j** of labelled fibroblasts, $n = 73$ (benign: $n = 31$, DCIS: $n = 16$, IDC: $n = 26$); Mann–Whitney test (**i**) or Kruskal–Wallis test followed by Dunn's multiple comparisons test (**j**). Data are presented as mean ± s.e.m; Mann–Whitney test. Source data are provided as a Source Data file

human protein networks, showed enrichment of inflammatory pathways, including NLRP3/IL-1β signalling, confirming the functional connectivity between genes of this pathway in breast cancer (Supplementary Fig. 2e). To analyse the expression of inflammasome genes at the protein level, we utilised a database of human IHC images[26], and found that NLRP3 expression in the stroma of breast tumours is elevated as compared with normal breast stroma (Fig. 2e). Quantitative analysis of fibroblast-specific staining indicated that NLRP3 is significantly upregulated in breast CAFs but not in normal breast fibroblasts (Fig. 2f, g). Furthermore, we analysed a cohort of 79 breast cancer patients and quantified the staining intensity and the abundance of IL-1β-expressing fibroblasts. These analyses indicated that IL-1β is significantly upregulated in CAFs in human breast carcinomas, in correlation with disease progression (Fig. 2h–j). Notably, as was previously demonstrated, inflammasome activation was also

apparent in tumour cells[27]. Taken together, these findings implicate the activation of the NLRP3 inflammasome pathway in CAFs in human breast carcinogenesis.

**Fibroblasts function as DAMP sensors via NLRP3 activation.** While DAMPs are known to be prominent activators of the NLRP3 inflammasome[28], fibroblasts were not previously implicated in sensing of tumour-derived DAMPs. Therefore, we next set out to determine whether DAMPs could trigger the activation of the NLRP3 inflammasome in mammary fibroblasts. NMFs isolated from mammary glands of non-transgenic mice were exposed to known DAMPs[28]: Adenosine triphosphate (ATP), monosodium urate (MSU) and $H_2O_2$. In addition, we analysed the effect of necrotic fluid extracted from advanced PyMT tumours, as it contains multiple DAMPs derived from necrotic

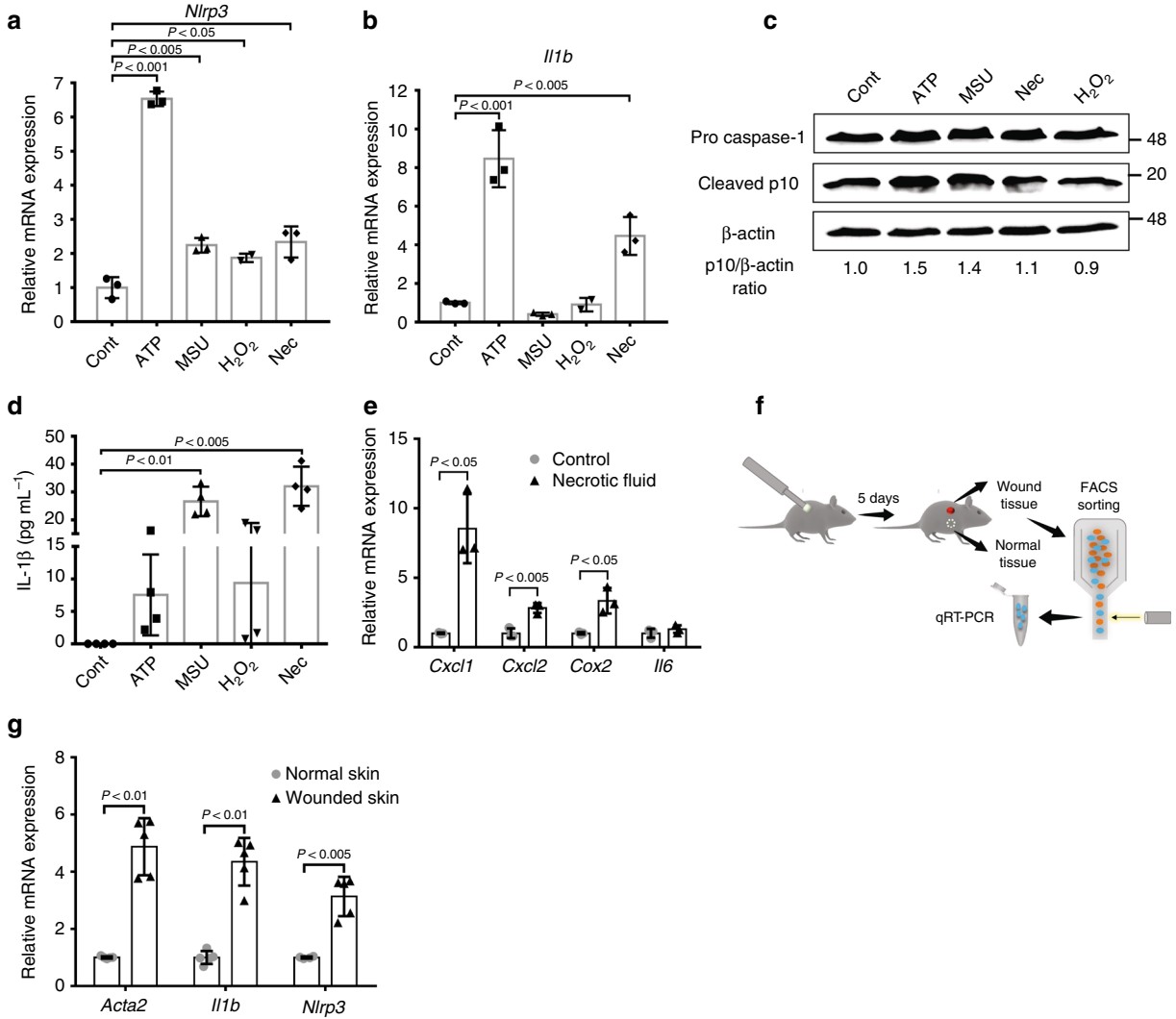

**Fig. 3** Fibroblasts function as DAMP sensors via NLRP3 inflammasome signalling. **a–e** Primary FVB/n NMFs were incubated for 24 h in control medium or in medium containing one of the following DAMPs: ATP 1 mM, $H_2O_2$ 300 μM, MSU 5 μg/mL, 5% necrotic fluid (extracted from late-stage PyMT tumours). *Nlrp3* **a** and *Il1b* **b** expression in fibroblasts were analysed by qRT-PCR. Data are presented as mean ± s.d.; One-way analysis of variance test followed by Dunnett's multiple comparisons test. Representative of three independent experiments. **c** Caspase-1 processing was assessed by western blot of cell lysates with anti-Caspase-1. β-actin was utilised as a loading control. The samples shown are derived from the same experiment. Data are representative of three independent experiments. **d** ELISA quantification of IL-1β secretion. Data are presented as mean ± s.d. of biological replicates ($n = 4$ per group); Kruskal–Wallis test followed by Dunn's multiple comparisons test. Representative of three independent experiments. **e** Primary FVB/n NMFs were incubated for 24 h with 5% necrotic fluid or control medium and expression of selected genes was analysed using qRT-PCR. Data are presented as mean ± s.d. of biological replicates ($n = 3$ per group); Welch's t-test. Representative of two independent experiments. **f** Schematic illustration of cutaneous wound assay. Cutaneous wounds were generated in 8 weeks old FVB/n mice using the dermal punch method. Wounded or control skin fibroblasts were isolated by FACS as PDGFRα+CD45−EpCAM− cells. **g** qRT-PCR analysis of the expression of selected genes in wound-derived fibroblasts as compared to their expression in fibroblasts from normal skin. $n =$ pool of 8 mice per group. Data are presented as mean ± s.d. of biological repeats; Welch's t-test. Representative of three independent biological experiments. Source data are provided as a Source Data file

cancer cells (Supplementary Fig. 3a, b). Multiple analyses indicated that fibroblasts can sense and respond to tissue damage signals: the expression of *Nlrp3* was significantly upregulated by all the tested DAMPs (Fig. 3a), and the expression of IL-1β was upregulated by ATP and necrotic fluid (Fig. 3b). Pro-caspase-1 was cleaved to its active form, manifested in the detection of the p10 subunit (Fig. 3c). We further assessed IL-1β secretion by ELISA, and found that all the different DAMPs tested and specifically tumour-derived necrotic fluid induced the secretion of IL-1β from fibroblasts (Fig. 3d). Moreover, we analysed the response of Gasdermin D, a component of the NLRP3 inflammasome pathway that is cleaved by active caspase-1[29]. We found that in response to incubation with DAMPs, the cleavage of

Gasdermin D in mammary fibroblasts was enhanced, further suggesting that fibroblasts respond to DAMPs by activation of the NLRP3 inflammasome pathway (Supplementary Fig. 3c). Interestingly, secretion of IL-1β was not directly linear with Casp-1 processing, suggesting that DAMPs also induce Casp-1-independent processing of IL-1β, as previously reported[30]. Notably, activation of pro-inflammatory signalling, previously shown to be a hallmark of CAFs[5,7], was not limited to the NLRP3 inflammasome: exposure of NMFs to DAMPs resulted in upregulation of a pro-inflammatory gene signature (Fig. 2e). Thus, tumour-related DAMPs can activate fibroblasts to functionally express the NLRP3 inflammasome and pro-inflammatory signalling.

Tumours often hijack physiological pathways to promote their growth. Fibroblasts are key sentinels of tissue integrity and their activation following tissue damage is central to the induction of inflammation and wound healing. Since tumours were compared to wounds that do not heal[31], we next asked whether DAMP sensing is a physiological characteristic of fibroblasts. To that end, we performed cutaneous wound assays and isolated fibroblasts derived from wounded skin or control skin (Fig. 3f). Gene expression analysis revealed upregulation of *Nlrp3* and *Il1b* in wound-associated fibroblasts, as well as upregulation of αSMA (*Acta2*), a marker of activated fibroblasts (Fig. 3g). These results suggest that sensing and responding to tissue damage signals via the NLRP3 inflammasome is a physiological feature of activated fibroblasts, instigated in CAFs by tumour-induced tissue damage.

**CAF-derived inflammasome signalling promotes tumour growth.** We next sought to investigate the functional role of CAF-derived NLRP3/IL-1β pathway in tumour growth in vivo. To that end, we performed orthotopic co-injections of fibroblasts and tumour cells into *Nlrp3*$^{-/-}$ mice. We utilised the syngeneic PyMT-derived mammary tumour cell line AT3 (C57BL/6 background)[32]. To specifically analyse the role of CAF-derived NLRP3, AT3 tumour cells were transduced with shRNA to silence *Nlrp3* expression (shNlrp3 AT3, Supplementary Fig. 4a) and injected to mammary glands of *Nlrp3*$^{-/-}$ mice admixed with *wild-type* (WT, C57BL/6) NMFs or *Nlrp3*$^{-/-}$ NMFs (Fig. 4a). Thus, the only source of NLRP3 in this experimental setting was the injected WT fibroblasts. Analysis of tumour growth revealed that tumours co-injected with *Nlrp3*$^{-/-}$ NMFs were significantly smaller than tumours injected with WT fibroblasts (Fig. 4b, c), suggesting that expression of NLRP3 in CAFs supports tumour growth. Seeking to characterise the mechanism by which CAF-derived NLRP3 affects tumour growth, we analysed parameters that were previously shown to be affected by CAFs, including fibrosis, immune cell recruitment, and angiogenesis[8]. Analysis of the results indicated no significant differences in collagen deposition (Sirius red staining) and blood vessel density (Meca-32) (Supplementary Fig. 4b, c). Interestingly, while FACS analysis of the infiltrating immune cells revealed no significant differences in infiltration of total myeloid cells (CD11b$^+$), macrophages (CD11b$^+$F480$^+$), or T cells (CD3$^+$) (Supplementary Fig. 4d–h), there was a significant reduction in recruitment of CD11b$^+$Gr1$^+$ myeloid cells into tumours co-injected with *Nlrp3*$^{-/-}$ NMFs as compared with WT fibroblasts (Fig. 4d). These results were confirmed by immunostaining of tumour tissue sections (Fig. 4e, f). Further analysis of the CD11b$^+$Gr1$^+$-infiltrating cells demonstrated that depletion of NLRP3 in CAFs-attenuated recruitment of the monocytic fraction of CD11b$^+$Gr1$^+$ cells (CD11b$^+$Ly6C$^{high}$Ly6G$^-$) into mammary tumours (Fig. 4g, h). Thus, expression of NLRP3 in CAFs is important for tumour growth, and mediates the recruitment of CD11b$^+$Gr1$^+$ myeloid cells into the tumour microenvironment.

In order to further elucidate the contribution of CAF-derived NLRP3/IL-1β pathway to tumorigenesis in the MMTV-PyMT model, we utilised Met-1 cells, a PyMT-derived cell line syngeneic with the FVB/n background[33]. The expression of IL-1β or NLRP3 was knocked down in mammary fibroblasts by two different clones of shRNA for *Il1b* or *Nlrp3* (shIl1b, shNlrp3 NMFs: Supplementary Fig. 5a, b). Met-1 cells were orthotopically injected alone, or were admixed and co-injected with control mammary fibroblasts (shScramble NMFs), or with fibroblasts depleted of *Il1b/Nlrp3* expression (Fig. 4i). Strikingly, while control mammary fibroblasts augmented tumour growth, genetic depletion of *Nlrp3* or *Il1b* in injected fibroblasts significantly attenuated tumour growth and weight (Fig. 4j, k), suggesting that

CAF-derived inflammasome is functionally important for tumour growth in vivo. Notably, depletion of *Nlrp3* in fibroblasts resulted in attenuated expression of *Il1b* and vice versa (Supplementary Fig. 4c, d).

Similarly to tumours injected in *Nlrp3*$^{-/-}$ mice, we observed no significant differences in collagen deposition (not shown), or the recruitment of total myeloid cells (CD11b$^+$), and macrophages (CD11b$^+$F480$^+$), and a mild increase in CD3$^+$ T cells (Supplementary Fig. 5e–i). However, in agreement with the experiments performed in *Nlrp3*$^{-/-}$ mice, FACS analysis of immune cell infiltration into Met-1 tumours revealed that genetic depletion of inflammasome signalling in CAFs resulted in attenuated recruitment of CD11b$^+$Gr1$^+$ myeloid cells (Fig. 4l). Interestingly, additional analysis of the CD11b$^+$Gr1$^+$ infiltrating cells demonstrated that while in the AT3 tumours injected in *Nlrp3*$^{-/-}$ mice depletion of inflammasome in CAFs resulted in reduced infiltration of the monocytic fraction of CD11b$^+$Gr1$^+$ cells, depletion of *Il1b* or *Nlrp3* in CAFs in Met-1 tumours injected to WT mice attenuated recruitment of the granulocytic (CD11b$^+$Ly6G$^+$Ly6C$^{low}$) fraction of CD11b$^+$Gr1$^+$ cells (Fig. 4m, n). These differences in the effects of inflammasome depletion in CAFs suggest that the distinct genetic backgrounds of mice may affect the composition of immune cells in tumours.

We next hypothesised that CAF-derived IL-1β may affect the recruitment of myeloid cells into the tumour microenvironment via activation of NF-κB-related signalling, as previously shown[5,34]. To test this hypothesis, mammary tumours were analysed for the expression of NF-κB downstream genes, known to be operative in the recruitment of myeloid cells, including *Ccl3, Cxcl2, Il6, SPP1, IL-23* and *TNFα*[35–38]. We found that tumours co-injected with CAFs in which *Nlrp3* or *Il1b* were depleted had significantly downregulated expression of pro-inflammatory genes related to immune cell recruitment (Fig. 4o–r, Supplementary Fig. 5j,k). Taken together, these findings demonstrate a tumour-promoting role for CAF-derived NLRP3/IL-1β, associated with alterations of immune cell infiltration into mammary tumours.

**CAF-derived IL-1β facilitates lung metastasis.** To investigate whether CAF-derived NLRP3 inflammasome is functional in other breast carcinoma models, we performed orthotopic co-injections with 4T1 cells, a highly metastatic breast cancer cell line, which models human triple negative breast cancer[39]. 4T1 cells were transduced with shRNA to knockdown the expression of IL-1β (Supplementary Fig. 6a), admixed with wild-type mammary fibroblasts or with *ll1b*$^{-/-}$ fibroblasts isolated from mammary glands of *ll1b*$^{-/-}$ mice, and injected into mammary glands of *ll1b*$^{-/-}$ recipients (Fig. 5a). In this experimental setting, WT fibroblasts are the only source of IL-1β. Analysis of tumours indicated that while tumour growth was only slightly inhibited in tumours injected with *ll1b*$^{-/-}$ fibroblasts (Fig. 5b), the tumour weights in this group were significantly smaller at end-stage as compared with tumours injected with WT fibroblasts (Fig. 5c). Moreover, analysis of lungs revealed a significant decrease in metastatic lesions in mice injected with *Il1b*$^{-/-}$ fibroblasts (Fig. 5d, e), suggesting that CAFs at the primary tumour affect the formation of distant metastases. Interestingly, metastatic lesions forming in these experiments were micro-metastases that could not be detected by CT scanning. We hypothesised that the total depletion of host IL-1β in the recipient mice may have hindered macro-metastases formation. Indeed, when we performed these experiments using WT mice as tumour recipients, 4T1 tumour cells formed spontaneous macro-metastases in lungs. Importantly, although primary tumours grew at similar rates (Supplementary Fig. 6b), mice injected with

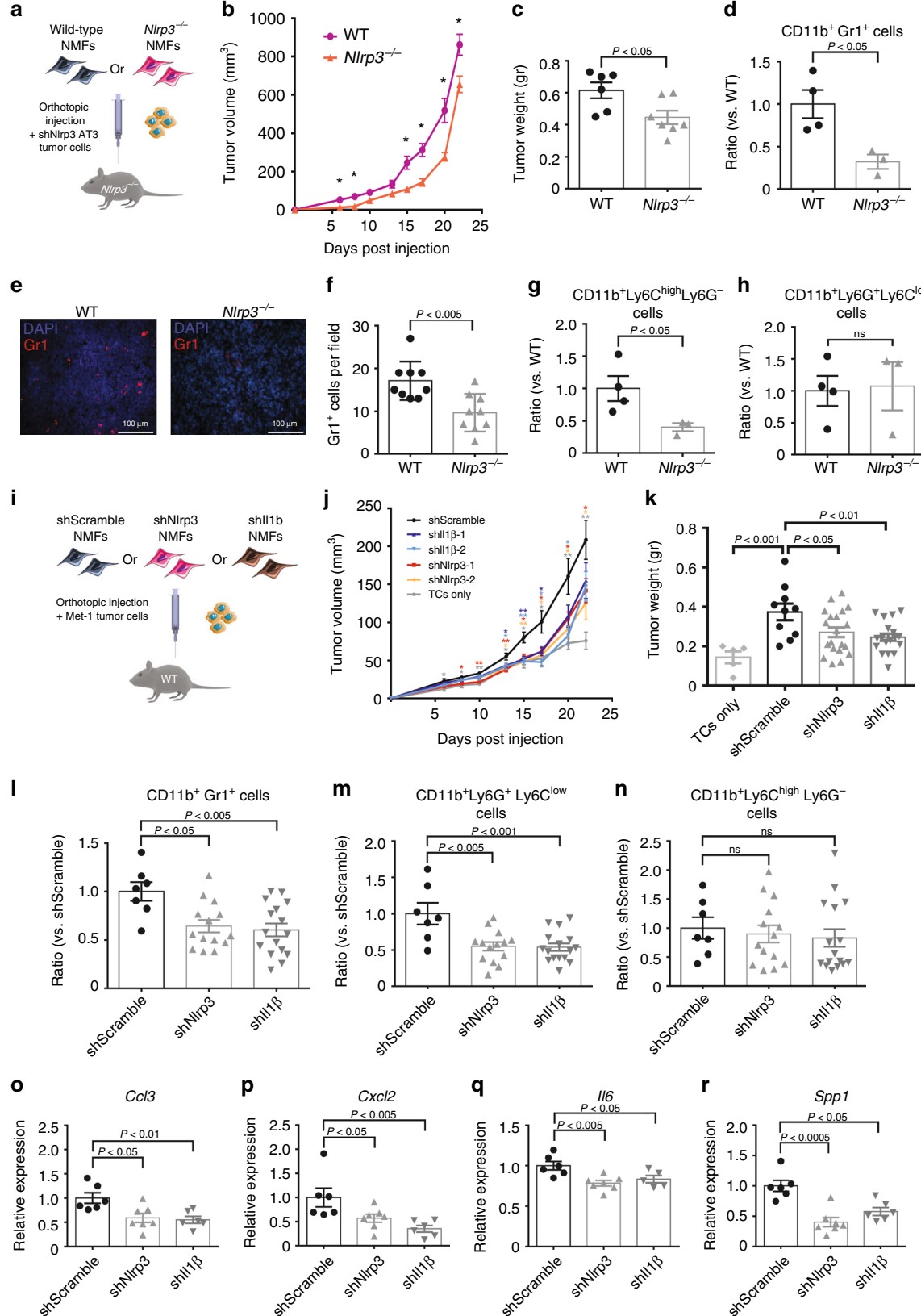

*Il1b*$^{-/-}$ fibroblasts had a significantly smaller lung metastatic burden compared to mice injected with WT fibroblasts (Fig. 5f–h), indicating that CAF-derived IL-1β contributes to breast cancer metastasis even in an IL-1β proficient microenvironment.

We therefore set out to elucidate the mechanism by which CAF-derived IL-1β at the primary tumour site enhances metastases formation. Initially, we analysed the immune cell milieu in primary tumours as well as in lung metastases. Similarly to the PyMT-derived models, the most significant changes

**Fig. 4** CAF-derived NLRP3 inflammasome signalling facilitates tumour growth. **a** Scheme of experiments analysed in **b–h**. shNlrp3 AT3 cells were orthotopically co-injected with WT ($Nlrp3^{+/+}$) or with $Nlrp3^{-/-}$ NMFs (C57BL/6) into C57BL/6 mice. **b** Growth curves of AT3 tumours. $n = 7$ mice per group. **c** Tumour weights at termination of experiment. $n = 6$ and 7 mice per group (WT and $Nlrp3^{-/-}$, respectively). **d** Flow cytometry analysis of $Gr1^+$ cell infiltration into AT3 tumours co-injected with WT NMFs ($n = 4$) or with $Nlrp3^{-/-}$ NMFs ($n = 3$). Data presented are percentage of $CD45^+$ cells, normalised to WT. **e** Representative images of staining with anti-Gr1 antibody in AT3 tumours. Scale bar, 100 μm. **f** Quantification of staining shown in **d**. Three fields of 3 mice/group were analysed; Data are presented as mean $Gr1^+$ cells/field ± s.d; Mann–Whitney test. **g, h** Flow cytometry analysis of $CD11b^+Ly6C^{high}Ly6G^-$ and $CD11b^+Ly6G^+Ly6C^-$ cell infiltration into AT3 tumours co-injected with WT NMFs ($n = 4$) or with $Nlrp3^{-/-}$ NMFs ($n = 3$). Data presented are percentage of $CD45^+$ cells, normalised to WT. **i** Scheme of experiments analysed in **j–r**. Met-1 cells were orthotopically co-injected with NMFs depleted of $Nlrp3$ or $Il1b$ expression (shNlrp3, and shIl1b), or with control NMFs (shScramble). **j** Growth curve and **k** weight at endpoint of Met-1 tumours. $n = 5$, 10, 20, 19 tumours (TCs only, shScramble, shNlrp3, and shIl1b, respectively). **l–n** Flow cytometry analysis of $Gr1^+$ cell infiltration into Met-1 tumours. $n = 7$, 14, and 16 per group (shScramble, shNlrp3, and shIl1b, respectively). Data presented are percentage of $CD45^+$ $CD11b^+$ cells, normalised to shScramble. **o–r** Met-1 tumours were analysed for the expression of selected genes using qRT-PCR. $n = 6$, 7, and 5–6 tumours per group (shScramble, shNlrp3, and shIl1b, respectively). Data are presented as fold change from shScramble. In **b–d, g, h, j, k**, data are presented as mean ± s.e.m; Welch's $t$-test, $*p < 0.05$, $**p < 0.005$. In **j–r** data are presented as mean ± s.e.m; One-way analysis of variance test followed by Tukey's multiple comparisons test. Data are representative of three independent biological experiments. Source data are provided as a Source Data file

induced by CAF-derived IL-1β were in the recruitment of $CD11b^+$ $Gr1^+$ cells. Interestingly, while in the primary tumour knockout of CAF-derived IL-1β resulted in attenuated recruitment of the granulocytic fraction ($CD11b^+Ly6G^+Ly6C^{low}$), in the metastatic microenvironment CAF-derived IL-1β was important for the recruitment of the monocytic fraction ($CD11b^+Ly6C^{high}Ly6G^-$) (Fig. 5i–l, Supplementary Fig. 6c–k). These results are in agreement with our findings in the PyMT-derived models, confirming that distinct genetic backgrounds and organ micro-environments drive specific compositions of the functional immune cells recruited.

Since the enhanced recruitment of $CD11b^+Gr1^+$ myeloid cells was consistent across all the breast cancer models we studied, we next set out to functionally characterise these cells. To that end, we isolated the granulocytic ($CD11b^+Ly6G^+Ly6C^{low}$) and monocytic ($CD11b^+Ly6C^{high}Ly6G^-$) fractions of $CD11b^+Gr1^+$ cells from mammary tumours and lung metastases. Gene expression analysis indicated that the $CD11b^+Gr1^+$ cells isolated from mammary tumours exhibited a typical gene signature of MDSCs, including upregulation of Arginase, iNOS and PD-L1. Notably, this was a tumour-specific phenotype, as $CD11b^+Gr1^+$ cells isolated from the spleen of the same mice did not have an immunosuppressive gene expression profile (Supplementary Fig. 7a–f). Moreover, T cell suppression assay with tumour-isolated $CD11b^+Gr1^+$ cells confirmed their immunosuppressive activity, suggesting that they are MDSCs (Supplementary Fig. 7m).

Seeking to get further mechanistic insight on the role of CAF-derived IL-1β in facilitating metastatic dissemination, we investigated its effect on tumour ECs. IL-1β is a key mediator of inflammation and exerts its pro-inflammatory functions, in part, by up-regulating the expression of adhesion molecules in ECs[40,41], thus enabling immune cell extravasation into inflamed tissues. In vitro analysis indicated that conditioned medium (CM) from DAMP-activated WT mammary fibroblasts but not from DAMP-activated $Il1b^{-/-}$ fibroblasts, upregulated the expression of $Selp$ (Selectin P) and $Vcam1$ in ECs (Fig. 6a, b). Importantly, these adhesion molecules were shown to facilitate cancer cell intravasation and extravasation by enhancing adhesion and transmigration of cancer cells through EC monolayers[42–44]. Moreover, trans-endothelial migration of breast cancer cells was attenuated when ECs were incubated with CM of $Il1b^{-/-}$ mammary fibroblasts as compared with CM from WT fibroblasts (Fig. 6c, d). We therefore hypothesised that IL-1β secretion from activated mammary CAFs upregulated the expression of adhesion molecules on tumour ECs in vivo, and thus facilitated cancer cell migration across the blood vessel endothelium. To test this hypothesis, we isolated ECs ($CD31^+CD45^-EpCAM^-$) from

mammary tumours and lung metastases and analysed the expression of adhesion molecules. Analysis of the results revealed that genetic depletion of $Il1b$ in fibroblasts in the primary tumour led to a significant decrease in the expression of E-Selectin, P-Selectin, ICAM-1 and VCAM-1 and in ECs sorted from primary tumours (Fig. 6e). Strikingly, the expression of adhesion molecules on lung ECs isolated from tumour-bearing mice was also attenuated in mice injected with $Il1b^{-/-}$ fibroblasts (Fig. 6f), suggesting that CAF-derived IL-1β has systemic effects that may support metastasis.

IL-1β was previously shown to promote tumour cell invasion and metastasis by modulating the expression of matrix metallo-proteinases in cancer cells, specifically MMP1, MMP3, and MMP10[45]. Therefore, we analysed whether CAF-derived IL-1β has a role in affecting tumour cell invasiveness. Indeed, we found that 4T1 cells co-injected with $Il1b^{-/-}$ fibroblasts expressed lower levels of $Mmp1$ and $Mmp3$ compared to cells co-injected with WT fibroblasts (Fig. 6g), suggesting that in addition to its effect on immune cell recruitment and on ECs, CAF-derived IL-1β enhances the invasiveness of breast cancer cells by upregulating their expression of MMPs.

Taken together, these findings suggest that CAF-secreted IL-1β facilitates breast cancer metastasis to lungs by multiple routes, including modulating the immune cell milieu at the metastatic site, and by upregulating the expression of adhesion molecules at the primary tumour, as well as the metastatic site, which may enhance tumour cell invasion and dissemination.

## Discussion
In this study, we uncovered a functional role for CAFs that links tissue damage with tumour-promoting inflammation. We showed that activation of the NLRP3 inflammasome pathway in CAFs, a physiological sensing mechanism of tissue damage, is hijacked in breast cancer to facilitate tumour progression and metastasis. Moreover, our data suggest that the mechanism by which CAF-derived NLRP3/IL-1β facilitate tumour growth and metastasis is by altering the immune cell milieu towards a tumour-tolerating phenotype, and by enhancing endothelial cell adhesiveness and cancer cell invasiveness (Fig. 7).

Activation of the NLRP3 inflammasome in fibroblasts was previously shown in pathologies associated with tissue damage and necrosis that trigger an intense inflammatory response, such as myocardial dysfunction[18,22,23], oral lesions[24], and fibrosis[19]. However, while fibroblasts are known to be mediators of tumour-promoting inflammation in cancer[6], they were not previously implicated in sensing tissue damage by pattern recognition via the inflammasome. We found that NMFs can respond to known DAMPs by upregulating the expression of

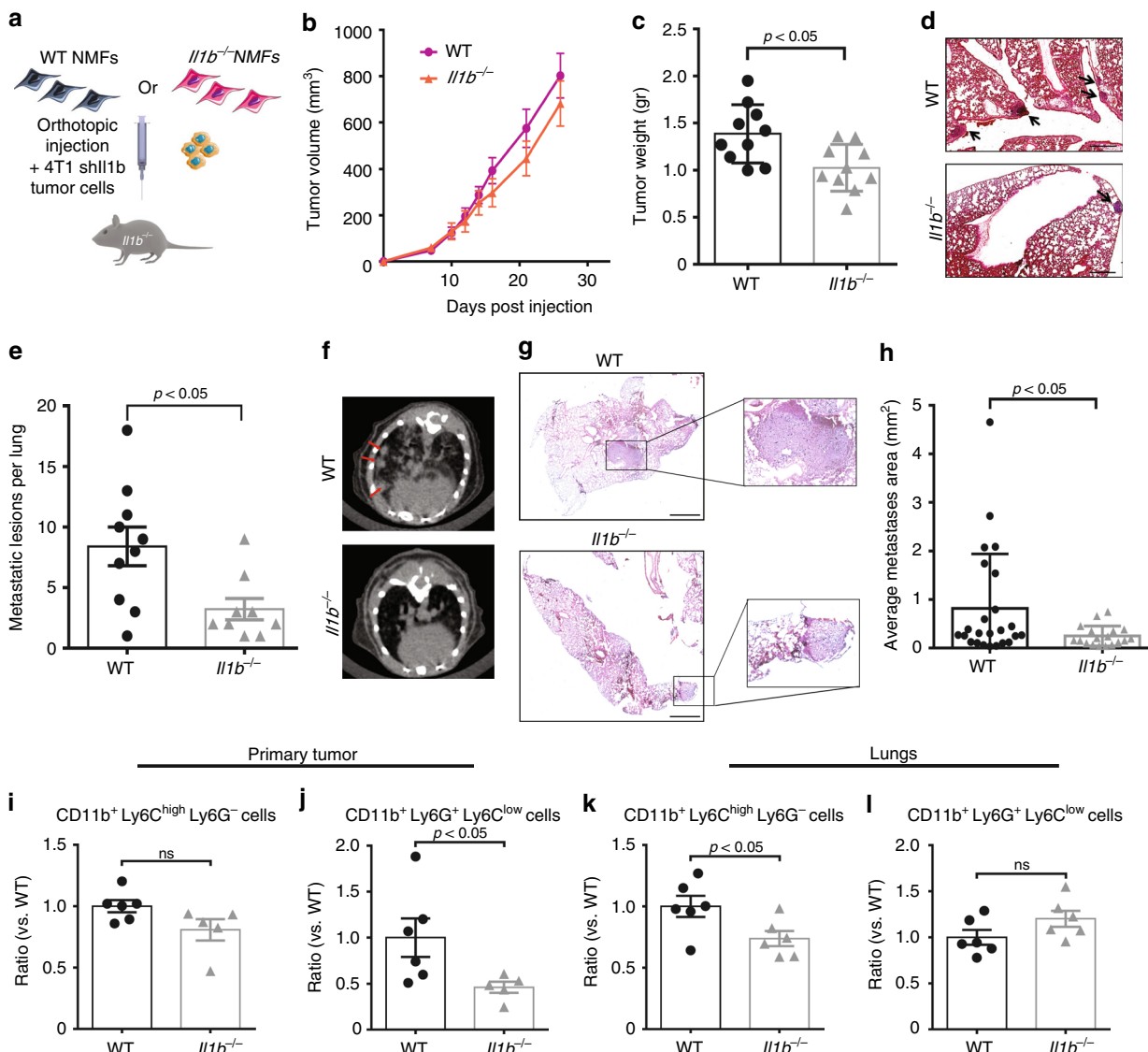

**Fig. 5** CAF-derived IL-1β facilitates lung metastasis. **a** Scheme of experiments analysed in **b–e**. shIl1b 4T1 mammary carcinoma cells were orthotopically co-injected with WT NMFs (*Il1b*+/+) or with *Il1b*−/− NMFs into the right inguinal mammary glands of *Il1b*−/− female mice. **b** Growth curves of 4T1 tumours injected with WT NMFs (*Il1b*+/+) or with *Il1b*−/− NMFs. n = 10 mice per group. **c** Tumour weight at termination of experiment. n = 10 mice per group. **d** Representative H&E staining of lung sections from injected mice. Insets depict spontaneous metastases (arrows). **e** Quantification of metastatic lesions per lung (serial sections of whole lungs were analysed). n = 9 and 10 individual mice per group (WT and *Il1b*−/−, respectively). Data are representative of three independent experiments. **f–h** BALB/c female mice were orthotopically injected with shIl1b 4T1 cancer cells admixed with WT NMFs (*Il1b*+/+) or with *Il1b*−/− NMFs. **f** Micro-CT intravital imaging of lungs shows spontaneous macrometastases (arrows). **g** H&E staining of lung tissue sections showing metastatic nodules. Insets depict enlarged fields of spontaneous metastases. **h** Quantification of metastatic load performed by analysing the area of metastatic foci per section of serially sectioned lungs. n = 6 individual mice per group. Representative of two independent experiments. **i–l** Flow cytometry analysis of MDSC infiltration into primary tumours (n = 5 or 6 individual tumours per group, WT and *Il1b*−/−, respectively) **i**, **j**, or lungs (n = 6 per group) **k**, **l** of 4T1-tumour bearing mice. Data presented are percentage of CD45+ CD11b+ cells, normalised to WT. Representative of three independent expirements. In **b**, **c**, **e**, **h**, **i**, **j**, **k**, **l** data are presented as mean ± s.e.m; Welch's *t*-test. Source data are provided as a Source Data file

*Nlrp3*, cleavage of pro-caspase-1 to its active form, and secretion of IL-1β. Importantly, we show that tumour-derived necrotic fluid was among the most potent inducers of the inflammasome in fibroblasts, further implicating tumour-associated tissue damage as a trigger of the inflammasome in CAFs. Moreover, our findings that fibroblasts in cutaneous wounds activate the inflammasome pathway suggest that sensing of DAMPs is a physiological feature of CAFs, as part of their role as sentinels of tissue damage and mediators of repair. In tumours, the chronic activation of these pathways results in sustained inflammation, and further tissue damage.

The NF-κB pathway, which regulates the transcription of IL-1β, was previously shown to be upregulated in CAFs and to mediate their pro-inflammatory functions in multiple cancers[5,46]. Here we expand these observations and demonstrate that the NLRP3 inflammasome pathway, which regulates the maturation and secretion of IL-1β is activated in CAFs in response to tissue damage, implicating a mechanism by which CAFs mediate tumour-promoting inflammation. Moreover, we show that physiological sensing of tissue damage by fibroblasts results in activation of the NLRP3 inflammasome, as well as an NF-κB-related pro-inflammatory gene signature, thus integrating both pathways.

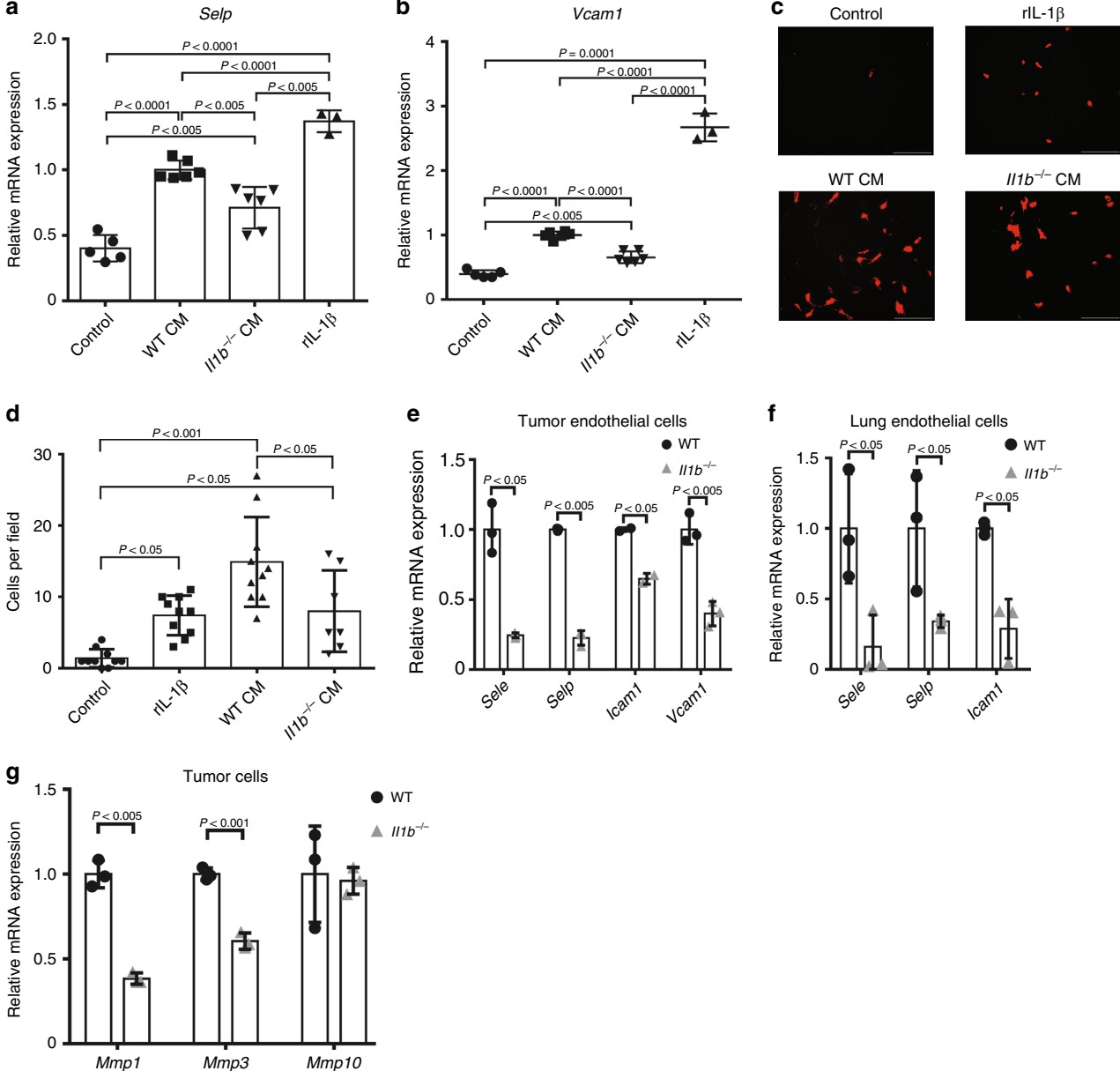

**Fig. 6** CAF-derived IL-1β upregulates expression of EC adhesion molecules and invasiveness markers. **a, b** C166 endothelial cells were incubated for 24 h in control medium, medium containing recombinant IL-1β (5 ng/mL), conditioned medium of MSU-treated WT mammary fibroblasts (Balb/C), or conditioned medium of MSU-treated *Il1b*⁻/⁻ fibroblasts. The expression of genes involved in tumour-cell intravasation/extravasation was analysed using qRT-PCR. Representative of three independent experiments. **c, d** Trans-endothelial migration of mCherry-labelled Met-1 cells supplemented with recombinant IL-1β (5 ng/mL), conditioned medium of WT NMFs, conditioned medium of *Il1b*⁻/⁻ NMFs, or control medium. **c** Representative images of trans-migrated Met1-mcherry cells. Scale bar, 100 μm. **d** Quantification of migrating cells per field of view. Representative of two independent experiments. **e, f** qRT-PCR analysis of the expression of adhesion and migration-related genes in endothelial cells (CD31⁺CD45⁻EpCAM⁻) isolated by FACS from **e** primary 4T1 tumours or **f** lungs of tumour-bearing mice. $n$ = pool of six mice per group. Representative of two independent biological experiments. **g** qRT-PCR analysis of the expression of invasion-related genes in tumour cells (EpCAM⁺CD45⁻) isolated by FACS from 4T1 mammary tumours. $n$ = pool of six tumours per group. Results are representative of two independent experiments. In **a, b, d** data are presented as mean ± s.d. of biological repeats; One-way analysis of variance test followed by Tukey's multiple comparisons test. In **e–g** data are presented as mean ± s.d. of technical repeats; Welch's *t*-test. Source data are provided as a Source Data file

Furthermore, expression analysis of NF-κB-regulated genes in tumours in which NLRP3 or IL-1β were manipulated in fibroblasts, indicated that the inflammasome pathway in CAFs is important for tumour expression of multiple cytokines and chemokines related to the recruitment of immune cells, including MDSCs. Thus, sensing of tissue damage by CAFs results in instigation of pro-inflammatory pathways in mammary tumours,

contributing to the formation of an immunosuppressed microenvironment.

Interestingly, *Il18*, which is also processed by the NLRP3 inflammasome, was downregulated in our transcriptome analysis of mammary CAFs. This is in agreement with the pro-tumourigenic effects we showed for CAF-derived inflammasome signalling, since IL-18 was previously suggested to have

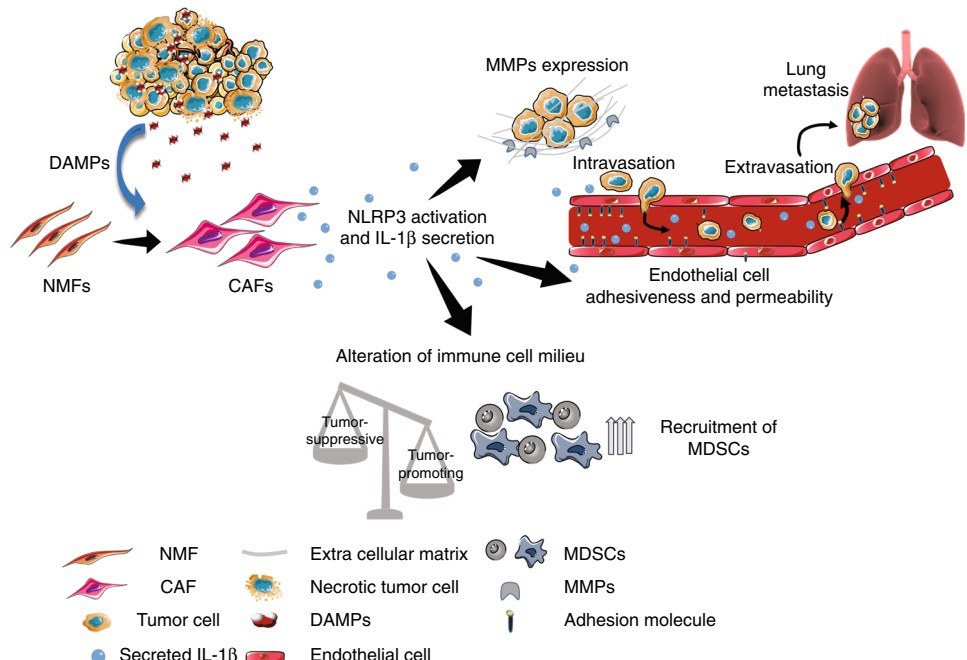

**Fig. 7** Summary. Activation of the NLRP3 inflammasome in cancer-associated fibroblasts links tissue damage with tumour-promoting inflammation in breast cancer progression and metastasis. Figure preparation partially involved elements from the Servier Medical Art. Figure preparation involved using elements from the Servier Medical Art and somersault18:24 (somersault1824.com)

anti-tumourigenic functions[47]. The balance between IL-1β and IL-18 may be cell-type, tissue, and context dependent, providing a possible explanation for the controversial role of the NLRP3 inflammasome in cancer.

By specifically targeting the expression of NLRP3 and IL-1β in mammary fibroblasts, we showed in multiple mouse models that CAF-derived inflammasome is functionally important for facilitating tumour growth. Importantly, while the delay in tumour growth was not striking, it was nevertheless consistent and significant, even in a setting of wild type recipients, in which NLRP3/IL-1β in the host microenvironment are intact, further emphasising the functional importance of CAF-derived inflammasome in supporting tumour growth.

The expression of IL-1β is elevated in a variety of cancers, including melanoma and breast cancer[27,48,49]. Moreover, mammary tumour growth was attenuated in $Nlrp3^{-/-}$ and $Casp1^{-/-}$ mice[50], implicating the NLRP3 inflammasome in facilitating breast cancer. Our data showing that fibroblasts can function as sensors of tissue damage, combined with specifically targeting the inflammasome pathway in fibroblasts demonstrate that activation of IL-1β in CAFs forms a functional link between tumour-associated tissue damage and CAF-mediated tumour-promoting inflammation.

Notably, we have elucidated the role of CAF-derived inflammasome signalling by specifically targeting the expression of NLRP3 and IL-1β in mammary fibroblasts. While beyond the scope of the current study, future generation of mice in which IL-1β will be conditionally ablated in fibroblasts will enable temporal dissection of its role during tumour progression and metastasis.

Importantly, in addition to promoting tumour growth at the primary tumour site, we found that depletion of CAF-derived IL-1β resulted in inhibition of lung metastasis. IL-1β was previously implicated in promoting metastasis of breast cancer to bone[51] and in supporting lung metastasis of melanoma and breast cancer, via recruitment and activation of γ/δ T cells[52,53]. Other studies suggested that IL-1β affects the mesenchymal transition and invasive behaviour of tumour cells[27,54]. Our findings suggest a

fibroblast-driven mechanism wherby IL-1β secreted from mammary fibroblasts, affects the expression of adhesion molecules on ECs at the primary tumour site, as well as in the lungs, thus facilitating the extravasation of tumour cells. Moreover, while, IL-1β knockout in fibroblasts at the primary tumour in the aggressive 4T1 triple-negative model was not sufficient to delay primary tumour growth, it was nevertheless associated with reduced infiltration of MDSCs in lungs, suggesting that signalling from the primary tumours has systemic effects on immune cell recruitment at the metastatic microenvironment. Moreover, CAF-derived IL-1β also upregulated the expression of metalloproteases in tumour cells, potentially enhancing their invasiveness. Thus, the physiological pro-inflammatory functions of IL-1β are activated in breast CAFs and facilitate tumour metastasis.

Analysis of immune cell infiltration in tumours injected with WT vs. inflammasome-deficient fibroblasts indicated that attenuated tumour growth and metastasis in mice in which NLRP3/IL-1β were knocked down, was associated with reduced recruitment of CD11b+Gr1+ myeloid cells. Interestingly, while this finding was consistent across all the models and genetic backgrounds studied, the changes in recruitment of the granulocytic (Ly6G+Ly6Clow) or monocytic (Ly6C+ Ly6G−) subpopulations to mammary tumours were distinct, depending on the organ (mammary gland vs. lungs) and on the model in which CAFs were manipulated. This observation is consistent with a recent study showing that genetic composition of tumours affects the immune landscape[55]. Moreover, these findings suggest that functional mechanisms in the tumour microenvironment are both plastic and robust—while the specific subpopulations of recruited cells may be distinct, depending on the cancer type, organ, or genetic background, cancer-associated immune cells perform similar tumour-promoting tasks.

Importantly, we analysed CD11b+Gr1+ cells isolated from tumours and found that they exhibited a typical gene signature of MDSCs, and were functional in suppressing T cell proliferation, confirming their immunosuppressive activity. Thus, CAF-derived IL-1β drives an immunosuppressive phenotype. CAFs were

previously reported in several studies to drive an immunosuppressive microenvironment in tumours by affecting the reprogramming of tumour-associated macrophages to an M2-like phenotype via secretion of Chi3L1[9], restricting the accumulation of T cells in tumours[56,57], modulating T cell immunity towards Th1 function[58], and by direct antigen-dependent killing of CD8+ T cells[59]. Our findings implicate activation of the NLRP3 inflammasome in CAFs in promoting the recruitment of MDSCs to breast tumours and to lung metastases, thus supporting the formation of an immunosuppressed microenvironment.

Our data suggest that targeting CAF-derived NLRP3/IL-1β pathway may be a beneficial therapeutic approach for breast cancer treatment. However, the function of the NLRP3/IL-1β pathway in tumourigenesis and metastasis was previously documented in immune cells with contradicting observations in different cancer types[28,60–64]. Targeting IL-1β was suggested to be a promising approach in cancer therapeutics[65], and our data suggest that this approach may be beneficial in blocking breast cancer metastasis. Nevertheless, taken together with recent reports that the effect of targeting CAFs may depend on tumour type and on the experimental systems that were used[46,66], future studies should be performed to carefully examine the benefit of targeting CAF-derived IL-1β in breast cancer. The indecisive role of the NLRP3 inflammasome in cancer is yet another reminder that complexity and context-dependent functions are hallmarks of the tumour microenvironment, and should therefore be carefully considered when designing therapeutic strategies.

In summary, our study reveals tumour-promoting functions of CAFs in sensing tumour-induced tissue damage, leading to activation of the NLRP3 inflammasome pathway and secretion of IL-1β, which promotes tumour progression and lung metastasis. These findings encourage the design of pre-clinical and clinical studies to examine the benefits of targeting the inflammasome pathway in breast cancer, which may be effective in blocking metastatic relapse.

## Methods

**Mouse strains**. All animals were maintained within the Tel Aviv University Specific Pathogen Free (SPF) Facility. All Animal procedures included in the study were granted ethical approval by the Tel Aviv University Institutional Animal Care and Use Committee. Animals used in experiments were not randomised, and the investigators were not blinded to allocation during experiments and outcome analysis. No statistical method was used to pre-determine sample size, but the sample size was chosen to be adequate to receive significant results as determined by preliminary experiments. Mice that died for unknown reasons were excluded from analysis. The FVB/N-Tg MMTV-PyMT 634Mul/J transgenic mice were a gift from Dr. Lisa Coussens. Heterozygote MMTV-PyMT mice are routinely bred with non-transgenic FVB/n females. Non-transgenic littermates were used as normal controls. The $Nlrp3^{-/-}$ (B6N.129-Nlrp3tm2Hhf/J) mice were purchased from Jackson Laboratory. Non-transgenic FVB/n mice, Balb/c mice, and C57BL/6J mice were purchased from Harlan, Israel. Mice were used for experiments at 6–9 weeks of age, unless otherwise stated.

**Cell lines**. Met-1 cells were a gift from Dr. Jeffrey Pollard. The 4T1 and AT3 cell lines were obtained from the laboratory of Dr. Scott Abrams, and received from Dr. Zvi Granot. C166 cells were purchased from ATCC (ATCC® CRL-2581™). All cell lines were routinely tested for mycoplasma using the EZ-PCR-Mycoplasma test kit (Biological Industries; 20-700-20). The cell lines were not authenticated.

To create shIl1b Met-1 cells, shIl1b 4T1 cells, and shNlrp3 AT3 cells, cells were infected with lentivirus-expressing shRNA for mouse Il1b (OpenBiosystems, RMM4534-EG16176) or Nlrp3 (OpenBiosystems, RMM4534-EG216799), or with control vector containing a scrambled shRNA sequence (OpenBiosystems, RMM4534-EG13131).

**Primary mammary fibroblasts**. NMFs were prepared from mammary glands isolated from 8 weeks old female mice. Mammary tissues were harvested, washed in PBS, minced thoroughly with scissors, and incubated for 30 min with DMEM supplemented with 0.1% collagenase II (Worthington, LS4176) and 0.1% collagenase IV (Worthington, LS004177) on stir plate in 37 °C water bath[21]. Cells were seeded on six-well plates that were pre-coated with rat-tail collagen (Corning; 354236) and grown in DMEM media supplemented with 10% FCS (Biological

industries; 01-055-1A, 04-001-1A), 1% penicillin–streptomycin (Biological industries; 03-031-1B) and 1% sodium-pyruvate (Biological industries; 03-042-1B), and maintained at 37 °C with 5% CO$_2$. All experiments were performed with low passage (p2-4) fibroblasts.

Preparation of shIl1b NMFs and shScramble NMFs was achieved by infecting the cells with lentivirus-expressing shRNA for mouse Il1b or with control vector containing a scrambled shRNA sequence (OpenBiosystems).

**Viral infections**. Viral production was carried out using PolyJet-mediated transfection (SignaGen Laboratories) of HEK293T according to manufacturer's instructions. Virus was concentrated by ultracentrifugation, and added to cells with polybrene (8 μg/mL). Infected cells were selected using Puromycin (3 μg/mL).

**Conditioned media**. Preparation of fibroblast-conditioned media (CM) was performed by incubating 80% confluent fibroblasts grown on collagen-coated 100 mm plastic dishes with 10 mL serum-free DMEM for 24 h. The medium was collected, filtered through 20 μm filters, flash-frozen in liquid nitrogen, and stored at −80 °C for later use.

**Transendothelial migration assay**. 6.5 mm Transwell plates with 8.0 μm pore polycarbonate membrane inserts (Sigma Aldrich, CLS3422) were coated with 10 μg/mL fibronectin in PBS for 12 h at 4 °C. C166 ECs ($2 × 10^5$) were trypsinized and suspended in control media, WT fibroblast CM, $Il1b^{-/-}$ fibroblast CM, or medium containing 10 ng/mL recombinant IL-1β, and seeded on the upper chamber for 8 h to allow the formation of a monolayer. Inserts were then washed in PBS, and mCherry-labelled Met-1 cells ($5 × 10^4$) were plated onto the upper chamber on top of the endothelial monolayer. Following incubation for 16 h, the upper side of the membrane was scraped gently with cotton swabs to remove non-invading cells. Membranes were fixed with 1% PFA and mounted on slides using DAPI Fluoromount-G (SouthernBiotech, 0100-20). Transendothelial migrated tumour cells were imaged using Leica DM4000B microscope and digital camera (Leica DFC 360FX), and quantified with ImageJ software. In all assays, analyses included five fields per insert. Experiments were performed in duplicates.

**Gene expression profiling**. RNA was purified from FACS sorted fibroblasts isolated from MMTV-PyMT tumours or mammary glands of FVB/n mice using Nucleuspin RNA II Kit (MACHERY-NAGEL). RNA concentration and purity was analysed using NanoDrop 2000c Spectrophotometer. RNA was converted to cDNA using iScriptTM reverse transcriptase (Bio-Rad). The cDNA was hybridised with Affymetrix GeneChip Mouse Gene 1.0 ST Arrays (Affymetrix, Santa Clara, CA, USA).

**Bioinformatics analysis**. Data analysis was performed on CEL files using Partek Genomics Suite v. 6.5 (Partek, St. Louis, MO, USA). Three replicates were used for all conditions except normal which was in duplicates. Data were normalised and summarised with the robust multi-average method followed by one-way analysis of variance (ANOVA)[67]. Genes of interest that were differentially expressed with $p < 0.05$ and fold-change difference cutoff > |1.5| were identified. Principal component analysis and hierarchical cluster analysis (Pearson's dissimilarity correlation and average linkage method) were performed using Partek Genomics Suite software. Functional analysis was performed using David and WebGestalt analysis platforms. Functional analysis of differentially expressed genes was performed using DAVID[68] and Gorilla[69] tools. Protein–protein interactions and function enrichment were obtained using STRING database[70].

**Quantitative real-time PCR**. Total RNA was isolated from cell pellets or cell monolayers using EZ-RNA II Kit according to manufacturer's instructions (Biological Industries, 20-410-100). RNA concentration and purity were analysed using NanoDrop 2000c spectrophotometer. cDNA synthesis was conducted using qScriptTM cDNA Synthesis Kit (Quanta Biosciences, 95047-025). Quantitative real-time PCRs (qRT-PCR) for mouse genes were conducted using SYBR® Green FastMIX® (Quanta Biosciences, P/N84071) in a StepOne™ Real-Time PCR System. All experiments were performed in triplicates. RQ ($2^{-\Delta Ct}$) was calculated. Relative expression was normalised to GAPDH. All primers and oligonucleotide sequences used are shown in Supplementary Table 1.

**Western blots**. Lysates were prepared from cell pellets in RIPA lysis buffer (Sigma-Aldrich, R0278). Protein determination was performed with Pierce BCA Protein Assay Kit (Thermo Fisher Scientific, 23227). Proteins from lysates were separated on Bis–Tris 12.5% acrylamide gels. Gels were transferred to nitrocellulose membranes (BioRad) and probed with rabbit anti-mouse Caspase-1 antibody (Santa Cruz, sc-514, 1 μg/ml, diluted 1:200) in 4 °C overnight, followed by incubation with HRP-conjugated secondary antibody (Jackson ImmunoResearch Laboratories, 111-035-144/062, 80 ng/ml) for 2 h at room temperature. Following membrane stripping, membranes were incubated with anti-mouse Gasdermin D (Abcam, ab209845, diluted 1:500) or with anti-mouse β-actin (Cell Signalling CST#4967, diluted 1:2000).

**Enzyme-linked immunosorbent assay (ELISA)**. To detect secreted IL-1β in conditioned media of cultured fibroblasts, the media was concentrated using Amicon Ultra-15 Centrifugal Filter Units with Ultracel-10 membrane (Merck Millipore, UFC901024) and ELISA was performed using Mouse IL-1 beta/IL-1F2 Quantikine ELISA Kit, according to manufacturer's instructions (R&D Systems, MLB00C).

**Detection of DAMPs in necrotic fluid**. Necrotic fluid was aspirated out of late-stage PyMT tumours and centrifuged at $2000 \times g$ for 10 min. Assessment of hydrogen peroxide or uric acid content was performed using detection kits (Bio-Vision, K265, K608).

**Orthotopic tumours transplantations**. Orthotopic intra-mammary injections: tumour cells ($1 \times 10^5$ Met-1 cells, AT3 cells, or 4T1 cells) were mixed with primary mammary fibroblasts ($3 \times 10^5$ cells) and then admixed 1:1 with matrigel (BD Biosciences, 354230), and injected into the nipple of the right inguinal mammary glands of 8 weeks old female mice. Tumours were measured every 2–3 days with a calliper, and tumour volumes were calculated using the formula $X^2 \times Y \times 0.52$ ($X =$ smaller diameter, $Y =$ larger diameter).

**Excisional wound-healing assay**. The dorsal skin of 8 weeks old FVB/n mice was shaved and cleaned with 70% ethanol under anaesthesia, and the skin was incised up to the level of the subcutaneous adipose tissue with a 5 mm Acu-Punch biopsy tool. Five days later, the wounded skin and control skin from the same animal were excised and digested into single cell suspension[5]. Briefly, wounded and control skin were washed in PBS, minced thoroughly with scissors, and incubated for 15 min with DMEM supplemented with 0.1% collagenase II (Worthington, LS4176) and 0.1% collagenase IV (Worthington, LS004177) on stir plate in 37 °C water bath.

**Mouse samples tissue preparation**. Mammary glands, tumours, and lungs were shortly washed in PBS. To prepare formalin-fixed paraffin-embedded (FFPE) blocks, tissues were incubated for 8–12 h in 10% formalin (Electron Microscopy Sciences) and transferred through ascending dilutions of ethanol before embedment in paraffin. In order to prepare fresh-frozen blocks, tissues were embedded in Optimal Cutting Temperature compound (O.C.T, Tissue-Tek) on dry ice. Serial sections were obtained to ensure equal sampling of the examined specimens (5–10 μm trimming).

**Quantification of metastatic load at the lungs**. Quantification of lung metastatic load was performed by analysing the number of metastatic nodules per section or by evaluating the metastatic area per section using ImageJ. Tissue sections (10 μm) were obtained by serial cutting of lungs. At least 20 sections per lung were analysed. Results are represented as the average number of metastatic nodules per lung (at least 9 mice/group) or as the average metastatic area per lung (10 mice/group).

**Immunohistochemistry**. Paraffin-embedded tissue sections were deparaffinized, rehydrated, and incubated with peroxidase-blocking solution (Dako, S2001) for 10 min to quench endogenous peroxidase activity. Antigen retrieval was carried out by microwave heating in a low pH target retrieval solution (Dako) for 15 min. Permeabilization was achieved by incubation with 0.25% Triton-X in PBS for 10 min. Tissue sections were incubated overnight at 4 °C with anti-NLRP3 antibody (Proteintech, 19771-1-AP, diluted 1:250), followed by incubation with biotinylated secondary antibody for 2 h at room temperature (Jackson ImmunoResearch laboratories; 711-065-152, diluted 1:400). Staining was developed using 3,3′-dia-minobenzidine (Sigma, D4168) and counterstained with Mayer Hematoxylin (Electron Microscopy Sciences). Images were obtained at ×20 or ×40 magnification using Leica DM4000B microscope and digital camera (Leica DFC295).

**Immunofluorescence**. Tissue sections were incubated overnight at 4 °C with the following anti-mouse antibodies: Ly-6G/Ly-6C (Gr-1)-APC/Cy7 (Biolegend, 108424), Meca32 (BD, 550563). For Meca32 staining an Alexa Fluor 488-conjugated Donkey Anti-Rat (Jackson ImmunoResearch laboratories; 712-546-153) was applied for 1 h at RT. Sections were washed with acidified water, and mounted with DAPI Fluoromount-G (Southern Biotech; 0100-20). Images were obtained at ×20 or ×40 magnification using Leica DM4000B microscope and digital camera (Leica DFC 360FX). Brightness and contrast were adjusted equally in all images presented.

**Fluorescence-activated cell sorting (FACS) for microarrays analysis and validation**. Mammary tumours isolated from MMTV-PyMT female mice or normal mammary tissue isolated from 8 weeks old FVB/n female mice were dis-associated into single cell suspensions and stained with anti-CD140a-PE (e-Bioscience, 12-1401-81, diluted 1:50) and anti-F4/80-FITC (Cederlane, CL8940F diluted 1:100). DAPI (Molecular probes; D3571, diluted 1:2000) was used to exclude dead cells. For validations, the single cell suspensions were incubated with anti-CD140a-PE, anti-CD45-FITC (eBioscience, 11-0451-81, diluted 1:200), and anti-EpCAM-APC (eBioscience, 17-5791-80, diluted 1:100), and fibroblasts were

isolated as $CD140a^+$ ($PDGFR\alpha^+$) $CD45^-$ $EpCAM^-$ cells. DAPI was used to exclude dead cells. Sorting was performed using BD FACS Aria II.

**FACS of ECs or myeloid cells**. Orthotopic 4T1 tumours or lungs of 4T1-tumour-bearing mice were isolated. Disassociation of tumours was performed as described[21]. For dissociation of lung tissue, lungs were minced into small pieces and incubated for 45 min with DMEM supplemented with 0.1% collagenase II (Worthington, LS4176) and 0.1% Dispase II (Sigma-Aldrich, 4942078001). Single cell suspensions were incubated with anti-mouse CD16/CD32 (eBioscience, 16-0161-82, diluted 1:50) for 15 min to reduce non-specific staining, followed by staining with the following anti-mouse antibodies: CD31 (PECAM-1)-FITC (eBioscience,11-0311-82, diluted 1:50), CD45-PE/Cy7 (eBioscience, 25-0451-82, diluted 1:200), and EpCAM-APC (eBioscience, 17-5791-80, diluted 1:100). ECs were isolated as $CD31^+CD45^-EpCAM^-$ cells. For myeloid cell isolation, tissues were disassociated into single cells suspensions as described above, and stained with the following anti-mouse antibodies: CD45-PE/Cy7 (eBioscience, 25-0451-82, diluted 1:200), CD11b-PerCP/Cy5.5 (eBioscience; 45-0112-82, diluted 1:100), Ly-6G-APC (Biolegend, 127614, dilu-ted 1:200) Ly-6C-FITC (Biolegend, 128006, diluted 1:200). DAPI was used to exclude dead cells. Sorting was performed using BD FACS Aria II.

**Intracellular staining of NLRP3**. Mammary tumours isolated from MMTV-PyMT female mice or normal mammary tissue isolated from 8 weeks old FVB/n female mice were disassociated into single cell suspensions and stained with anti CD140a-PE (eBioscience, 12-1401-81, diluted 1:50), anti CD45-FITC (eBioscience, 11-0451-81, diluted 1:100), and anti EpCAM-PE/Cy7 (Biolegend, 118216, diluted 1:100). The intracellular staining for NLRP3 was performed with APC-conjugated anti-body (R&D, IC7578A, diluted 1:10) following fixation and permeabilization of cells using BD Cytofix/Cytoperm kit (BD Bioscience, 554714) according to the manu-facturer's protocol.

**Flow cytometry analysis of immune cells in injected tumours**. Orthotopic Met-1 or AT3 tumours were isolated, dissociated into single cell suspension, and incubated for 15 min with anti-mouse CD16/CD32 to reduce non-specific staining. Infiltration of immune cells into tumours was analysed by staining with the following anti-mouse antibodies: CD45-PE/Cy7 (eBioscience, 25-0451-82, diluted 1:200), CD45-BV570 (Biolegend, 103136, diluted 1:100), CD3e-FITC (eBioscience, 14-0031-82, diluted 1:100), CD4-APC/Cy7 (Biolegend, 100414, diluted 1:100), anti-CD8a-APC (eBioscience, 17-0081-82, diluted 1:100), CD11b-PerCP/Cy5.5 (eBioscience; 45-0112-82, diluted 1:100), F4/80-APC (eBioscience, 17-4801-82, diluted 1:50), Ly-6G/Ly-6C (Gr-1)-APC/Cy7 (Biolegend, 108424, diluted 1:100), Ly-6G-APC (Biolegend, 127614, diluted 1:200), Ly-6C-FITC (Biolegend, 128006, diluted 1:200), Ly-6C-BV605 (Biolegend, 128036, diluted 1:200). DAPI was used to exclude dead cells. Acquisition was performed with Beckman-Coulter Gallios flow cytometer and data analysis was done with FlowJo software (version X.0.7).

**T cell suppression assay**. Splenocytes were isolated from the spleen of $Il1b^{-/-}$ mice, and labelled with CFSE (5 M, Biolegend, 423801). $1 \times 10^5$ cells in 100 μL per well were plated in a 96-well plate pre-coated with anti-mouse CD3ε (1 μg/mL, SouthernBiotech, 1530-01) and anti-CD28 (5 μg/mL, SouthernBiotech, 1610-01).

$CD11b^+Gr1^+$ cells were isolated using FACS and added to the stimulated T cells ($5 \times 10^4$). Dilution of CFSE was evaluated 3 days later by flow cytometry.

**Stromal mRNA profiling of inflammasome-related genes**. Stromal gene expression-profiling data for 53 breast cancers were obtained from Gene Expression Omnibus (GEO) (GSE9014). Replicate samples were averaged prior to analysis.

**Human NLRP3 staining**. Publically available IHC staining (The Human Protein Atlas[26]) of breast cancer and normal tissues stained with anti-NLRP3 antibody were analysed (https://www.proteinatlas.org/ENSG00000162711-NLRP3/pathology/tissue/breast+ cancer#img, HPA012878 antibody). Breast cancer tissue specimens from surgical pathology included both ductal and lobular cancers. We analysed the expression of NLRP3 in fibroblasts only in the ductal subtype. For fibroblasts staining quantification, each image was divided into three fields and assessed by an expert pathologist (L.L.T.). Abundance was defined as the percen-tage of stained fibroblasts out of the total fibroblasts per field. Staining intensity of fibroblasts in each field was evaluated and rated between 1 and 3 (1—low, 2—intemediate, and 3—high). In total, two normal samples and eight IDC samples were analysed.

**Human IL-1β staining**. Tissue preparation and processing was performed by the Pathology Departments of the Tel Aviv Sourasky Medical Center and Meir Medical Center[27]. 5 μM-thick serial sections were prepared from paraffin blocks obtained at time of diagnosis. Heat-induced epitope retrieval was performed by immersing the sections in citrate buffer at 95–100 °C for 22 min. Sections were blocked with a solution containing 0.05% sodium azide and 0.05% Tween 20 in PBS. Next, the sections were incubated with anti-human IL-1β antibody (Exalpha, Watertown,

MA; Cat# L140M) for 1 h at room temperature, followed by incubation with a biotinylated secondary antibody for 10 min. Benign, DCIS, and IDC tissue sections stained for IL-1β were examined by an expert pathologist (L.L.T.) for the staining intensity of fibroblasts and the abundance of stained fibroblasts in the tissue and rated between 1 and 3 (1—low, 2—intemediate, and 3—high). Thirty-three benign samples, 19 DCIS samples, and 30 IDC samples. From the IDC samples analysed, 25 were from the luminal subtype (distinction between luminal-A and luminal-B subtypes was not available), and five were triple negative. Twenty-three of the IDC patients were treated with chemotherapy (1), radiotherapy (16), or both (6). Statistical significance between group means was calculated using Kruskal–Wallis test followed by Dunn's multiple comparisons test.

**Statistical analysis**. Statistical analyses were performed using GraphPad Prism software. Standard parametric two-tailed $t$-tests were applied and standard errors were calculated for each data set with $n > 2$. For multiple comparisons of parametric data, the one-way ANOVA test followed by Tukey's multiple comparisons test were used. Significance between non-parametric data was calculated using the Kruskal–Wallis test. $p$-value of < 0.05 was considered statistically significant for all data sets. Bar graphs represent mean and standard deviation (SD) or mean and standard error of the mean (SEM) across multiple independent experimental repeats or across technical repeats, as stated.

**Reporting summary**. Further information on research design is available in the Nature Research Reporting Summary linked to this article.

## Data availability

The microarray data in support of this study have been deposited in the Gene Expression Omnibus (GEO) database under the accession code GSE129189. The source data underlying Figs. 1–6 and Supplementary Figs. 2–7 are provided as a Source Data file. All the other data supporting the findings of this study are available within the article and its supplementary information files and from the corresponding author upon reasonable request. A reporting summary for this article is available as a Supplementary Information file.

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

## Acknowledgements

This research was supported by grants to N.E. from the European Research Council (ERC) under the European Union's Horizon 2020 research and innovation programme (grant agreement No. 637069 MetCAF), from the Israel Science Foundation (#813/12), the Israel Cancer Association, and The Israel Cancer Research Fund (*Research Career Development Award and Project Grant*). We thank Ms. Hadar Cohen for her help with the revision process.

## Author contributions

N. Ershaid, Y.S., and N.E. (Neta Erez) conceived the study and designed the experiments. N. Ershaid, Y.S., H.D., Y.R., O.S., N.C., L.M., and A.B.-S. performed the experiments. L.L.-T. and M.P.S analysed data. M.Y., A.B.-B., M.G., and R.A. provided essential reagents and expertise. N. Ershaid and N.E. wrote the manuscript. N.E. supervised the study.

## Competing interests

The authors declare no competing interests.
