## [Peer Review File · Nature Communications]

Reviewers' comments:

Reviewer #4 (Remarks to the Author):

I agree with previous 3 reviewers that this manuscript is interesting, novel, and of significance the breast cancer field, but at technical level, it indeed needs to be improved.

1) Fig 2e-j: Original reviewer 1 pointed out the quality of human protein atlas pictures. It is really not about the resolution since the scanned pictures can be high enough for publication. It is about the quality of the antibody. Most Human Protein Atlas antibodies are Sigma polyclonal, in particular NLRP3 (HPA012878) is assigned uncertain for quality. It requires stringent quality control in order to publish (a monoclonal antibody should be preferred for IHC data, or using a siRNA or CRISPR KO NLRP3 in human cells to show its specificity). This particular true when a cytoplasmic protein ends up in nuclear staining (panel 2e, IDC) of essentially all carcinoma cells.

2) Fig 1g has a similar problem. NLRP3 is very intensive from adipose tissue (g, top panel normal) as well as cancer cells (g, lower panel tumor). A NLRP3 KO control mammary tissues should be included.

3) Fig 3c, Fig S2, and the new rebuttal western blotting images versus IL-1beta production data (Fig. 1d) are very inconsistent. I assume all data came from the normal mammary fibroblasts from FVB/N mice. Control group has very high Casp-1 processing (lane 1, 3c), but no IL-1beta processing at all; Nec group has similar Casp-1 processing (lane 4, 3c), but highest IL-1beta production.

Along the same line, ATP has the highest priming (mRNA of Nlrp3, Il-1beta, and Casp-1 processing, but very low in IL-1beta production.

This almost indicates IL-1beta processing is not totally dependent on Casp-1 processing and activation. The authors should provide discussion or data to address the apparent inconsistency.

Reviewer #5 (Remarks to the Author):

Here I summarize the comments from the Reviewers and replies from authors and my thoughts about the comments, as requested by the Editor:

"(b) in response to the comments about the quality of the images in Figure 2e-j I think it is an insufficient response to say they come from a commercial website. Poor quality images are poor quality images no matter where they come from"

I agree with authors that the images are good enough and IL-1b expressing cells can be quantified using these images

"that the effects on tumor growth remain modest and the authors have only provided the important control of injecting tumor cells alone in one of their models. Despite redoing the experiment in Fig. 4j to include tumor cells alone, the authors have again terminated the experiment when the tumors are quite small. Consequently, they still do not fully address the original question as to whether the effect is just a delay in growth. The authors argue that although the data "might not be striking, they are nevertheless statistically significant and consistent throughout multiple experiments and models". But this is not the case. No growth impairment was seen with the 4T1 cells in either WT or IL1b^{-/-} mice. part of the reason for bringing this up is that in their abstract the authors that that CAF driven inflammasome facilitated mammary carcinogenesis. And in their title they refer to tumor-promoting inflammation in breast cancer progression. I appreciated that they have demonstrated an effect on metastasis in the 4T1 model but if, as they claim in their rebuttal, that this is their most striking result"

This is a legit concern, especially that 4T1 cells on Sup. Fig 5 do not have a delay in growth but have delay in met's and changes in myeloid cells- which questions how uniform the mechanism is, but I guess authors have used their best available model and should get a pass on it if this will be more thoroughly discussed in the discussion section as a limitation of the study

Here is couple of my own comments:

Fig 4G vs 4M different subsets are recruited in complete knockouts vs fibroblast specific knockouts, indicating that the role of CAF Nlrp3/IL-1b system may be inferior to the ones in other cell types.

Again, no experiments are needed but this should be discussed

"Fig 6 CAF-derived IL-1 β upregulates trans-endothelial migration and invasiveness of breast cancer cell"

This reviewer thinks that the title is misleading. Experiments were not performed in vivo. In the system used, "any IL-1b" provided would cause the effect (see results with recombinant IL-1b). the fact that there is only a supernatant from CAF in the system in vitro does not prove that other sources of IL-1b are not much ore superior in vitro

As for the Reviewer 3, I agree with the authors that it will be problematic now to generate Fibroblastic Cre x IL-1b flox mice if these are not readily available although some labs have them. Contraty to what the Reviewer suggested, IL-1R floxed mouse is in no way a substitution for these studies, as authors do not study IL-1R signaling in fibroblasts here, only IL-1b release and effects on other cell types, including cancer cells.

Point-by-point response to Reviewers' comments:

Reviewer #4 (Remarks to the Author):

I agree with previous 3 reviewers that this manuscript is interesting, novel, and of significance the breast cancer field,

We thank the reviewer for acknowledging the novelty and importance of our work.

but at technical level, it indeed needs to be improved.

1) Fig 2e-j: Original reviewer 1 pointed out the quality of human protein atlas pictures. It is really not about the resolution since the scanned pictures can be high enough for publication. It is about the quality of the antibody. Most Human Protein Atlas antibodies are Sigma polyclonal, in particular NLRP3 (HPA012878) is assigned uncertain for quality. It requires stringent quality control in order to publish (a monoclonal antibody should be preferred for IHC data, or using a siRNA or CRISPR KO NLRP3 in human cells to show its specificity). This particular true when a cytoplasmic protein ends up in nuclear staining (panel 2e, IDC) of essentially all carcinoma cells.

We would like to clarify that our goal in including the images from the Human Protein Atlas was not to show expression of NLRP3 in carcinoma cells. NLRP3 expression in tumor cells is well-known (Reviewed in: Karki R. et al. Nature Rev. Cancer 2019, Moosavi, M. et al. Mol. Cancer 2018). Rather, we wanted to strengthen the novel findings shown in Fig. 2A, that the expression of NLRP3 is upregulated in the stromal compartments of human breast tumors, specifically in fibroblasts. We quantified all the available tissues in the database, and the differences in the staining were striking and unequivocal, as seen in the quantification shown in Fig. 2f, performed with a specialist pathologist.

While we agree with the reviewer that polyclonal antibodies are not ideal, had there been background or unspecific staining it would be evident in both normal and tumor tissues, which was not the case here. The controls suggested by the reviewer are interesting, however, we do not have access to the tissue sections in the Human Protein Atlas, and collecting new human samples would require a new IRB ethical approval, would realistically take many months, and is beyond the scope of the study at this stage. Moreover, a monoclonal antibody to NLRP3 for IHC is not available, to the best of our knowledge.

Specifically knocking down NLRP3 in human fibroblasts would also be a lengthy process, and would require the use of the same available polyclonal antibody.

Since this staining is supportive data, rather than our key finding, and the results we presented of fibroblast staining are clearly unequivocal, we respectfully think that additional controls with human tissues in this specific case are unlikely to yield a meaningful outcome.

2) Fig 1g has a similar problem. NLRP3 is very intensive from adipose tissue (g, top panel normal) as well as cancer cells (g, lower panel tumor). A NLRP3 KO control mammary tissues should be included.

To address this comment, we performed additional staining in mammary tissue from NLRP3 KO mice. The results confirmed that staining with anti-NLRP3 antibody used in the manuscript did not yield positive staining in *Nlrp3*^{-/-} tissues, including fibroblasts (Fig. A.). This is similar to what we have shown in FVB/n normal mammary glands (Fig. 1g, top panel), while in mammary tumors there was intense staining in both tumor cells (as reported in the literature) and in CAFs (Fig. 1g, lower panel).

Figure A

Representative IHC staining of NLRP3 in normal mammary glands of *Nlrp3*^{-/-} mice. Scale bar, 50µm.

3) Fig 3c, Fig S2, and the new rebuttal western blotting images versus IL-1beta production data (Fig. 1d) are very inconsistent. I assume all data came from the normal mammary fibroblasts from FVB/N mice. Control group has very high Casp-1 processing (lane 1, 3c), but no IL-1beta processing at all; Nec group has similar Casp-1 processing (lane 4, 3c), but highest IL-1beta production.

Along the same line, ATP has the highest priming (mRNA of *Nlrp3*, IL-1beta, and Casp-1 processing, but very low in IL-1beta production.

This almost indicates IL-1beta processing is not totally dependent on Casp-1 processing and activation. The authors should provide discussion or data to address the apparent inconsistency.

The reviewer is correct in their statement: IL-1 β was indeed demonstrated in various studies to be activated, processed and secreted via pro-inflammatory signaling pathways that are Casp-1 independent (e.g. Wan et al. Nature Comm. 2015, Lukens et al. PNAS 2014, Provoost et al. J. Immunology 2011, Kono et al. J. Immunology 2012). To address this point and to better discuss our findings presented in Fig. 3, we added accordingly a statement and reference to the revised manuscript (page 4). The revised text is: "Interestingly, secretion of IL-1 β was not directly linear with Casp-1 processing, suggesting that DAMPs also induce Casp-1- independent processing of IL-1 β , as previously reported ³⁰."

We thank the reviewer for bringing this to our attention.

Reviewer #5 (Remarks to the Author):

Here I summarize the comments from the Reviewers and replies from authors and my thoughts about the comments, as requested by the Editor:

"(b) in response to the comments about the quality of the images in Figure 2e-j I think it is an insufficient response to say they come from a commercial website. Poor quality images are poor quality images no matter where they come from"

I agree with authors that the images are good enough and IL-1b expressing cells can be quantified using these images.

We thank the reviewer for confirming our response, that the quality of the images presented is sufficient for the quantitative analysis that we provided.

"that the effects on tumor growth remain modest and the authors have only provided the important control of injecting tumor cells alone in one of their models. Despite redoing the experiment in Fig. 4j to include tumor cells alone, the authors have again terminated the experiment when the tumors are quite small. Consequently, they still do not fully address the original question as to whether the effect is just a delay in growth. The authors argue that although the data "might not be striking, they are nevertheless statistically significant and consistent throughout multiple experiments and models". But this is not the case. No growth impairment was seen with the 4T1 cells in either WT or IL1b^{-/-} mice. part of the reason for bringing this up is that in their abstract the authors that that CAF driven inflammasome facilitated mammary carcinogenesis. And in their title they refer to tumor-promoting inflammation in breast cancer progression. I appreciated that they have demonstrated an effect on metastasis in the 4T1 model but if, as they claim in their rebuttal, that this is their most striking result"

This is a legit concern, especially that 4T1 cells on Sup. Fig 5 do not have a delay in growth but have delay in met's and changes in myeloid cells- which questions how uniform the mechanism is, but I guess authors have used their best available model and should get a pass on it if this will be more thoroughly discussed in the discussion section as a limitation of the study.

To address this comment, we added a sentence in our discussion section that clearly states that in the 4T1 model, k/o of IL-1 β in fibroblasts did not inhibit primary tumor growth, but rather had an effect on lung metastasis (page 9). The revised paragraph reads: "**Moreover, while, IL-1 β knockout in fibroblasts at the primary tumour in the aggressive 4T1 triple-negative model was not sufficient to delay primary tumour growth**, it was nevertheless associated with reduced infiltration of MDSCs in lungs, suggesting that signalling from the primary tumours has systemic effects on immune cell recruitment at the metastatic microenvironment. Moreover, CAF-derived IL-1 β also upregulated the expression of metalloproteases in tumour cells, potentially enhancing their invasiveness. Thus, the physiological pro-inflammatory functions of IL-1 β are activated in breast CAFs and facilitate tumour metastasis".

Here is couple of my own comments:

Fig 4G vs 4M different subsets are recruited in complete knockouts vs fibroblast specific knockouts, indicating that the role of CAF Nlrp3/Il-1b system may be inferior to the ones in other cell types.

Again, no experiments are needed but this should be discussed.

This is an important comment that we have thoroughly discussed both in previous rebuttals and in our manuscript (page 9, Discussion section):

"Analysis of immune cell infiltration in tumours injected with WT vs. inflammasome-deficient fibroblasts indicated that attenuated tumour growth and metastasis in mice in which NLRP3/IL-1 β were knocked down, was associated with reduced recruitment of CD11b⁺Gr1⁺ myeloid cells. Interestingly, while this finding was consistent across all the models and genetic

backgrounds studied, **the changes in recruitment of the granulocytic (Ly6G⁺Ly6C^{low}) or monocytic (Ly6C⁺ Ly6G⁻) subpopulations to mammary tumours were distinct, depending on the organ (mammary gland vs. lungs) and on the model in which CAFs were manipulated.** This observation is consistent with a recent study showing that genetic composition of tumours affects the immune landscape ⁶⁰. Moreover, **these findings suggest that functional mechanisms in the tumour microenvironment are both plastic and robust- while the specific subpopulations of recruited cells may be distinct, depending on the cancer type, organ, or genetic background, cancer-associated immune cells perform similar tumour-promoting tasks.”**

"Fig 6 CAF-derived IL-1 β upregulates trans-endothelial migration and invasiveness of breast cancer cell"

This reviewer thinks that the title is misleading. Experiments were not performed *in vivo*. In the system used, “any IL-1b” provided would cause the effect (see results with recombinant IL-1b). the fact that there is only a supernatant from CAF in the system *in vitro* does not prove that other sources of IL-1b are not much more superior *in vitro*.

We are not sure what the reviewer meant by “Experiments were not performed *in vivo*”. The figure includes data from *in vivo* experiments (Fig. 6e-g), specifically indicating the role of CAF-derived IL-1 β (rather than “any IL-1b”) in affecting the expression of adhesion molecules on endothelial cells, and MMPs in tumor cells. Since these experiments were performed in IL-1 β knockout mice, the only source of IL-1 β was the WT fibroblasts, and thus the findings are directly correlated with CAF-derived IL-1 β . However, in response to this comment, and to more accurately reflect our findings, we changed the title of Fig. 6, as suggested. The new title is: “CAF-derived IL-1 β upregulates expression of endothelial adhesion molecules and invasion-related factors in breast cancer cells”.

We thank the reviewer for bringing to our attention that this was not sufficiently clear.

As for the Reviewer 3, I agree with the authors that it will be problematic now to generate Fibroblastic Cre x IL-1b flox mice if these are not readily available although some labs have them. Contrary to what the Reviewer suggested, IL-1R floxed mouse is in no way a substitution for these studies, as authors do not study IL-1R signaling in fibroblasts here, only IL-1b release and effects on other cell types, including cancer cells.

We thank the reviewer for confirming our claims.

REVIEWERS' COMMENTS:

Reviewer #4 (Remarks to the Author):

All concerns are addressed. Accepted.